# Convergence and Stability of Graph Convolutional Networks on Large Random Graphs

**Nicolas Keriven**[*]
CNRS, GIPSA-lab, Grenoble, France
nicolas.keriven@cnrs.fr

**Alberto Bietti**[*]
NYU Center for Data Science, New York, USA[†]
alberto.bietti@nyu.edu

**Samuel Vaiter**
CNRS, IMB, Dijon, France
samuel.vaiter@u-bourgogne.fr

## Abstract

We study properties of Graph Convolutional Networks (GCNs) by analyzing their behavior on standard models of random graphs, where nodes are represented by random latent variables and edges are drawn according to a similarity kernel. This allows us to overcome the difficulties of dealing with discrete notions such as isomorphisms on very large graphs, by considering instead more natural geometric aspects. We first study the convergence of GCNs to their continuous counterpart as the number of nodes grows. Our results are fully non-asymptotic and are valid for relatively sparse graphs with an average degree that grows logarithmically with the number of nodes. We then analyze the stability of GCNs to small deformations of the random graph model. In contrast to previous studies of stability in discrete settings, our continuous setup allows us to provide more intuitive deformation-based metrics for understanding stability, which have proven useful for explaining the success of convolutional representations on Euclidean domains.

## 1 Introduction

Graph Convolutional Networks (GCNs [9, 15, 26]) are deep architectures defined on graphs inspired by classical Convolutional Neural Networks (CNNs [29]). In the past few years, they have been successfully applied to, for instance, node clustering [11], semi-supervised learning [26], or graph regression [23, 20], and remain one of the most popular variant of Graph Neural Networks (GNN). We refer the reader to the review papers [7, 47] for more details.

Many recent results have improved the theoretical understanding of GNNs. While some architectures have been shown to be universal [36, 24] but not implementable in practice, several studies have characterized GNNs according to their power to distinguish (or not) graph *isomorphisms* [48, 12, 35] or compute combinatorial graph parameters [13]. However, such notions usually become moot for large graphs, which are almost never isomorphic to each other, but for which GCNs have proved to be successful in identifying large-scale structures nonetheless, *e.g.,* for segmentation or spectral clustering [11]. Under this light, a relevant notion is that of *stability*: since GCNs are trained then tested on different (large) graphs, how much does a change in the graph structure affect its predictions? In the context of signals defined on Euclidean domains, including images or audio, convolutional representations such as scattering transforms or certain CNN architectures have been shown to be stable to *spatial deformations* [34, 5, 40]. However the notion of deformations is not well-defined on

---

[*]Equal contribution.
[†]Work done while AB was at Inria Paris.

discrete graphs, and most stability studies for GCNs use purely discrete metrics that are less intuitive for capturing natural changes in structure [17, 19, 49].

In statistics and machine learning, there is a long history of modelling large graphs with random models, see for instance [6, 21, 27, 37] and references therein for reviews. *Latent space models* represent each node as a vector of latent variables and independently connect the nodes according to a *similarity kernel* applied to their latent representations. This large family of random graph models includes for instance Stochastic Block Models (SBM) [22], graphons [32], random geometric graphs [38], or $\varepsilon$-graphs [10], among many others [37]. A key parameter in such models is the so-called *sparsity factor* $\alpha_n$ that controls the number of edges in $\mathcal{O}(n^2 \alpha_n)$ with respect to the number of nodes $n$. The *dense* case $\alpha_n \sim 1$ is the easiest to analyze, but often not realistic for real-world graphs. On the contrary, many questions are still open in the *sparse* case $\alpha_n \sim 1/n$ [1]. A middle ground, which will be the setting for our analysis, is the so-called *relatively sparse* case $\alpha_n \sim \log n/n$, for which several non-trivial results are known [30, 25], while being more realistic than the dense case.

**Outline and contributions.** In this paper, we analyze the convergence and stability properties of GCNs on large random graphs. We define a "continuous" counterpart to discrete GCNs acting on graph models in Section 2, study notions of invariance and equivariance to isomorphism of random graph models, and give convergence results when the number of nodes grows in Section 3. In particular, our results are fully non-asymptotic, valid for relatively sparse random graphs, and unlike many studies [45, 41] we do not assume that the similarity kernel is smooth or bounded away from zero. In Section 4, we analyze the stability of GCNs to small deformation of the underlying random graph model. Similar to CNNs [34, 5], studying GCNs in the continuous world allows us to define intuitive notions of model deformations and characterize their stability. Interestingly, for GCNs equivariant to permutation, we relate existing discrete notions of distance between graph signals to a Wasserstein-type metric between the corresponding continuous representations, which to our knowledge did not appear in the literature before.

**Related work on large-scale random graphs.** There is an long history of studying the convergence of graph-related objects on large random graphs. A large body of works examine the convergence of the eigenstructures of the graph adjacency matrix or Laplacian in the context of spectral clustering [4, 45, 30, 43] or learning with operators [41]. The theory of graphons [32] defines (dense) graph limits for more general metrics, which is also shown to lead to spectral convergence [16]. Closer to our work, notions of Graph Signal Processing (GSP) such as the graph Fourier Transform have been extended to graphons [42] or sampling of general Laplacian operators [31]. Partial results on the capacity of GCNs to distinguish dense graphons are derived in [33], however their analysis based on random walks differs greatly from ours. In general, many of these studies are asymptotic [45, 42], valid only in the dense case [45, 41, 42, 31, 33], or assume kernels that are smooth or bounded away from zero [41], and thus exclude several important cases such as SBMs, $\varepsilon$-graphs, and non-dense graphs altogether. By specifying models of (relatively sparse) random graphs, we derive non-asymptotic, fully explicit bounds with relaxed hypotheses.

**Related work on stability.** The study of stability to deformations has been pioneered by Mallat [34] in the context of the scattering transform for signals on Euclidean domains such as images or audio signals [8, 2], and was later extended to more generic CNN architectures [5, 40]. A more recent line of work has studied stability properties of GCNs or scattering representations on discrete graphs, by considering certain well-chosen discrete perturbations and metrics [17–19, 49], which may however have limited interpretability without an underlying model. In contrast, our continuous setup allows us to define more intuitive geometric perturbations based on deformations of random graph models and to obtain deformation stability bounds that are similar to those on Euclidean domains [34]. The work [28] also studies deformation stability of GNNs, but considers architectures specifically constructed for mesh data, rather than general graphs. We note that [31] also considers GCN representations with continuous graph models, but the authors focus on the different notion of "transferability" of graph filters on different discretizations of the *same* underlying continuous graph structure, while we consider *explicit deformations* of this underlying structure and obtain non-asymptotic bounds for the resulting random graphs.

## 2 Preliminaries

**Notations.** The norm $\|\cdot\|$ is the Euclidean norm for vector and spectral norm for matrices, and $\|\cdot\|_F$ is the Frobenius norm. We denote by $\mathcal{B}(\mathcal{X})$ the space of bounded real-valued functions on $\mathcal{X}$ equipped with the norm $\|f\|_\infty = \sup_x |f(x)|$. Given a probability distribution $P$ on $\mathcal{X}$, we denote by $L^2(P)$ the Hilbert space of $P$-square-integrable functions endowed with its canonical inner product. For multivariate functions $f = [f_1, \ldots, f_d]$ and any norm $\|\cdot\|$, we define $\|f\| = (\sum_{i=1}^d \|f_i\|^2)^{\frac{1}{2}}$. For two probability distributions $P, Q$ on $\mathbb{R}^d$, we define the Wasserstein-2 distance $\mathcal{W}_2^2(P, Q) = \inf\{\mathbb{E}\|X - Y\|^2 \mid X \sim P, Y \sim Q\}$, where the infimum is over all joint distributions of $(X, Y)$. We denote by $f_\sharp P$ the push-forward of $P$ by $f$, that is, the distribution of $f(X)$ when $X \sim P$.

A graph $G = (A, Z)$ with $n$ nodes is represented by a symmetric adjacency matrix $A \in \{0,1\}^{n \times n}$ such that $a_{ij} = 1$ if there is an edge between nodes $i$ and $j$, and a matrix of signals over the nodes $Z \in \mathbb{R}^{n \times d_z}$, where $z_i \in \mathbb{R}^{d_z}$ is the vector signal at node $i$. We define the normalized Laplacian matrix as $L = L(A) = D(A)^{-\frac{1}{2}} A D(A)^{-\frac{1}{2}}$, where $D(A) = \mathrm{diag}(A1_n)$ is the degree matrix, and $(D(A)^{-\frac{1}{2}})_i = 0$ if $D(A)_i = 0$. The normalized Laplacian is often defined by $\mathrm{Id} - L$ in the literature, however this does not change the considered networks since the filters include a term of order $0$.

**Graph Convolutional Networks (GCN).** GCNs are defined by alternating filters on graph signals and non-linearities. We use analytic filters (said of order-$k$ if $\beta_\ell = 0$ for $\ell \geqslant k + 1$):

$$h : \mathbb{R} \to \mathbb{R}, \quad h(\lambda) = \sum_{k \geqslant 0} \beta_k \lambda^k. \tag{1}$$

We write $h(L) = \sum_k \beta_k L^k$, *i.e.,* we apply $h$ to the eigenvalues of $L$ when it is diagonalizable.

A GCN with $M$ layers is defined as follows. The signal at the input layer is $Z^{(0)} = Z$ with dimension $d_0 = d_z$ and columns $z_j^{(0)} \in \mathbb{R}^n$. Then, at layer $\ell$, the signal $Z^{(\ell)} \in \mathbb{R}^{n \times d_\ell}$ with columns $z_j^{(\ell)} \in \mathbb{R}^n$ is propagated as follows:

$$\forall j = 1, \ldots d_{\ell+1}, \quad z_j^{(\ell+1)} = \rho\left(\sum_{i=1}^{d_\ell} h_{ij}^{(\ell)}(L) z_i^{(\ell)} + b_j^{(\ell)} 1_n\right) \in \mathbb{R}^n, \tag{2}$$

where $h_{ij}^{(\ell)}(\lambda) = \sum_k \beta_{ijk}^{(\ell)} \lambda^k$ are learnable analytic filters, $b_j^{(\ell)} \in \mathbb{R}$ are learnable biases, and the activation function $\rho : \mathbb{R} \to \mathbb{R}$ is applied pointwise. Once the signal at the final layer $Z^{(M)}$ is obtained, the output of the entire GCN is either a signal over the nodes denoted by $\Phi_A(Z) \in \mathbb{R}^{n \times d_{out}}$ or a single vector denoted by $\bar{\Phi}_A(Z) \in \mathbb{R}^{d_{out}}$ obtained with an additional pooling over the nodes:

$$\Phi_A(Z) \overset{\text{def.}}{=} Z^{(M)} \theta + 1_n b^\top, \quad \bar{\Phi}_A(Z) \overset{\text{def.}}{=} \frac{1}{n} \sum_{i=1}^n \Phi_A(Z)_i, \tag{3}$$

where $\theta \in \mathbb{R}^{d_M \times d_{out}}$, $b \in \mathbb{R}^{d_{out}}$ are the final layer weights and bias, and $\Phi_A(Z)_i \in \mathbb{R}^{d_{out}}$ is the output signal at node $i$. This general model of GCN encompasses several models of the literature, including all spectral-based GCNs [9, 15], or GCNs with order-1 filters [26] which are assimilable to message-passing networks [20], see [47, 7] for reviews. For message-passing networks, note that almost all our results would also be valid by replacing the sum over neighbors by another aggregation function such as $\max$. We assume (true for ReLU, modulus, or sigmoid) that the function $\rho$ satisfies:

$$|\rho(x)| \leqslant |x|, \quad |\rho(x) - \rho(y)| \leqslant |x - y|. \tag{4}$$

Two graphs $G = (A, Z)$, $G' = (A', Z')$ are said to be *isomorphic* if one can be obtained from the other by relabelling the nodes. In other words, there exists a *permutation matrix* $\sigma \in \Sigma_n$, where $\Sigma_n$ is the set of all permutation matrices, such that $A = \sigma \cdot A' \overset{\text{def.}}{=} \sigma A' \sigma^\top$ and $Z = \sigma \cdot Z' \overset{\text{def.}}{=} \sigma Z'$, where "$\sigma \cdot$" is a common notation for permuted matrices or signal over nodes. In graph theory, functions that are *invariant* or *equivariant* to permutations are of primary importance (respectively, permuting the input graph does not change the output, or permutes the output). These properties are hard-coded in the structure of GCNs, as shown by the following proposition (proof in Appendix **??**).

**Proposition 1.** *We have* $\Phi_{\sigma \cdot A}(\sigma \cdot Z) = \sigma \cdot \Phi_A(Z)$ *and* $\bar{\Phi}_{\sigma \cdot A}(\sigma \cdot Z) = \bar{\Phi}_A(Z)$.

**Random graphs.** Let $(\mathcal{X}, d)$ be a compact metric space. In this paper, we consider latent space graph models where each node $i$ is represented by an unobserved latent variable $x_i \in \mathcal{X}$, and nodes

are connected randomly according to some *similarity kernel*. While the traditional graphon model [32] considers (without lost of generality) $\mathcal{X} = [0, 1]$, it is often more intuitive to allow general spaces to represent meaningful variables [14]. We consider that the observed signal $z_i \in \mathbb{R}^{d_z}$ is a function of the latent variable $x_i$, without noise for now. In details, a *random graph model* $\Gamma = (P, W, f)$ is represented by a probability distribution $P$ over $\mathcal{X}$, a symmetric kernel $W : \mathcal{X} \times \mathcal{X} \to [0, 1]$ and a bounded function $f : \mathcal{X} \to \mathbb{R}^{d_z}$. A random graph $G$ with $n$ nodes is then generated as follows:

$$\forall j < i \leqslant n : \quad x_i \overset{iid}{\sim} P, \quad z_i = f(x_i), \quad a_{ij} \sim \mathrm{Ber}(\alpha_n W(x_i, x_j)). \tag{5}$$

where Ber is the Bernoulli distribution. We define $d_{W,P} \overset{\text{def.}}{=} \int W(\cdot, x)dP(x)$ the *degree function* of $\Gamma$. As outlined in the introduction, the sparsity factor $\alpha_n \in [0, 1]$ plays a key role. The so-called *relatively sparse* case $\alpha_n \sim \frac{\log n}{n}$ will be the setting for our analysis.

Let us immediately make some assumptions that will hold throughout the paper. We denote by $N(\mathcal{X}, \varepsilon, d)$ the $\varepsilon$-covering numbers (that is, the minimal number of balls of radius $\varepsilon$ required to cover $\mathcal{X}$) of $\mathcal{X}$, and assume that they can be written under the form $N(\mathcal{X}, \varepsilon, d) \leqslant \varepsilon^{-d_x}$ for some constant $d_x > 0$ (called the *Minkowski* dimension of $\mathcal{X}$), and $\mathrm{diam}(\mathcal{X}) \leqslant 1$. Both conditions can be obtained by a rescaling of the metric $d$. Let $c_{\min}, c_{\max} > 0$ be constants. A function $f : \mathcal{X} \to \mathbb{R}$ is said to be $(c_{\text{Lip.}}, n_\mathcal{X})$-piecewise Lipschitz if there is a partition $\mathcal{X}_1, \ldots, \mathcal{X}_{n_\mathcal{X}}$ of $\mathcal{X}$ such that, for all $x, x'$ in the same $\mathcal{X}_i$, we have $|f(x) - f(x')| \leqslant c_{\text{Lip.}} d(x, x')$. All considered random graph models $\Gamma = (P, W, f)$ satisfy that for all $x \in \mathcal{X}$,

$$\|W(\cdot, x)\|_\infty \leqslant c_{\max}, \quad d_{W,P}(x) \geqslant c_{\min}, \quad W(\cdot, x) \text{ is } (c_{\text{Lip.}}, n_\mathcal{X})\text{-piecewise Lipschitz.} \tag{6}$$

Unlike other studies [45, 41], we do *not* assume that $W$ itself is bounded away from 0 or smooth, and thus include important cases such as SBMs (piecewise constant $W$) and $\varepsilon$-graphs (threshold kernels).

**Continuous GCNs.** Since $d_{W,P} > 0$, we define the normalized Laplacian operator $\mathcal{L}_{W,P}$ by

$$\mathcal{L}_{W,P}f \overset{\text{def.}}{=} \int \frac{W(\cdot, x)}{\sqrt{d_{W,P}(\cdot)d_{W,P}(x)}} f(x)dP(x). \tag{7}$$

Analytic filters on operators are $h(\mathcal{L}) = \sum_k \beta_k \mathcal{L}^k$, with $\mathcal{L}^k = \mathcal{L} \circ \ldots \circ \mathcal{L}$. We do *not* assume that the filters are of finite order (even if they usually are in practice [15]), however we will always assume that $\sum_k k |\beta_k| (2c_{\max}/c_{\min})^k$ converges. Similar to the discrete case, we define continuous GCNs (c-GCN) that act on random graph models, by replacing the input signal $Z$ with $f$, the Laplacian $L$ by $\mathcal{L}_{W,P}$, and propagating functions instead of node signals *i.e.,* we take $f^{(0)} = f$ the input function with coordinates $f_1^{(0)}, \ldots, f_{d_z}^{(0)}$ and:

$$\forall j = 1, \ldots, d_{\ell+1}, \quad f_j^{(\ell+1)} = \rho \circ \left( \sum_{i=1}^{d_\ell} h_{ij}^{(\ell)}(\mathcal{L}_{W,P})f_i^{(\ell)} + b_j^{(\ell)} 1(\cdot) \right), \tag{8}$$

where $1(\cdot)$ represent the constant function 1 on $\mathcal{X}$. Once the final layer function $f^{(M)} : \mathcal{X} \to \mathbb{R}^{d_M}$ is obtained, the output of the c-GCN is defined as in the discrete case, either as a multivariate function $\Phi_{W,P}(f) : \mathcal{X} \to \mathbb{R}^{d_{out}}$ or a single vector $\bar{\Phi}_{W,P}(f) \in \mathbb{R}^{d_{out}}$ obtained by pooling:

$$\Phi_{W,P}(f) \overset{\text{def.}}{=} \theta^\top f^{(M)} + b1(\cdot), \quad \bar{\Phi}_{W,P}(f) = \int \Phi_{W,P}(f)(x)dP(x). \tag{9}$$

Hence the *same* parameters $\{(\beta_{ijk}^{(\ell)}, b_j^{(\ell)})_{ijk\ell}, \theta, b\}$ define both a discrete and a continuous GCN, the latter being (generally) not implementable in practice but useful to analyze their discrete counterpart.

For a random graph model $\Gamma = (P, W, f)$, and any invertible map $\phi : \mathcal{X} \to \mathcal{X}$, we define $\phi \cdot W \overset{\text{def.}}{=} W(\phi(\cdot), \phi(\cdot))$ and $\phi \cdot f \overset{\text{def.}}{=} f \circ \phi$. Recalling that $(\phi^{-1})_\sharp P$ is the distribution of $\phi^{-1}(x)$ when $x \sim P$, it is easy to see that $\phi \cdot \Gamma \overset{\text{def.}}{=} ((\phi^{-1})_\sharp P, \phi \cdot W, \phi \cdot f)$ defines the same probability distribution as $\Gamma = (P, W, f)$ over discrete graphs. Therefore, we say that $\Gamma$ and $\phi \cdot \Gamma$ are *isomorphic*, which is a generalization of isomorphic graphons [32] when $P$ is the uniform measure on $[0, 1]$. Note that, technically, $\phi$ needs only be invertible on the support of $P$ for the above definitions to hold. As with discrete graphs, functions on random graph models can be invariant or equivariant, and c-GCNs satisfy these properties (proof in Appendix **??**).

**Proposition 2.** *For all $\phi$, $\Phi_{\phi \cdot W, (\phi^{-1})_\sharp P}(\phi \cdot f) = \phi \cdot \Phi_{W,P}(f)$ and $\bar{\Phi}_{\phi \cdot W, (\phi^{-1})_\sharp P}(\phi \cdot f) = \bar{\Phi}_{W,P}(f)$.*

In the rest of the paper, most notation-heavy multiplicative constants are given in the appendix. They depend on $c_{\min}, c_{\max}, c_{\text{Lip.}}$ and the operator norms of the matrices $B_k^{(\ell)} = (\beta_{ijk}^{(\ell)})_{ij}$.

# 3 Convergence of Graph Convolutional Networks

In this section, we show that a GCN applied to a random graph $G \sim \Gamma$ will be close to the corresponding c-GCN applied to $\Gamma$. In the invariant case, $\bar{\Phi}_A(Z)$ and $\bar{\Phi}_{W,P}(f)$ are both vectors in $\mathbb{R}^{d_{out}}$. In the equivariant case, we will show that the output signal $\Phi_A(Z)_i \in \mathbb{R}^{d_{out}}$ at each node is close to the function $\Phi_{W,P}(f)$ evaluated at $x_i$. To measure this, we consider the (square root of the) Mean Square Error at the node level: for a signal $Z \in \mathbb{R}^{n \times d_{out}}$, a function $f : \mathcal{X} \to \mathbb{R}^{d_{out}}$ and latent variables $X$, we define $\mathrm{MSE}_X (Z, f) \stackrel{\text{def.}}{=} (n^{-1} \sum_{i=1}^n \|Z_i - f(x_i)\|^2)^{1/2}$. In the following theorem we use the shorthand $D_{\mathcal{X}}(\rho) \stackrel{\text{def.}}{=} \frac{c_{\text{Lip.}}}{c_{\min}} \sqrt{d_x} + \frac{c_{\max} + c_{\text{Lip.}}}{c_{\min}} \sqrt{\log \frac{n_{\mathcal{X}}}{\rho}}$.

**Theorem 1** (Convergence to continuous GCN). *Let $\Phi$ be a GCN and $G$ be a graph with $n$ nodes generated from a model $\Gamma$, denote by $X$ its latent variables. There are two universal constants $c_1, c_2$ such that the following holds. Take any $\rho > 0$, assume $n$ is large enough such that $n \geqslant c_1 D_{\mathcal{X}}(\rho)^2 + \frac{1}{\rho}$, and the sparsity level is such that $\alpha_n \geqslant c_2 c_{\max} c_{\min}^{-2} \cdot n^{-1} \log n$. Then, with probability at least $1 - \rho$,*

$$\mathrm{MSE}_X (\Phi_A(Z), \Phi_{W,P}(f)) \leqslant R_n \stackrel{\text{def.}}{=} C_1 D_{\mathcal{X}} \left( \frac{\rho}{\sum_\ell d_\ell} \right) n^{-\frac{1}{2}} + C_2 (n\alpha_n)^{-\frac{1}{2}},$$

$$\left\| \bar{\Phi}_A(Z) - \bar{\Phi}_{W,P}(f) \right\| \leqslant R_n + C_3 \sqrt{\log(1/\rho)} n^{-\frac{1}{2}}.$$

**Discussion.** The constants $C_i$ are of the form $C_i' \|f\|_\infty + C_i''$ and detailed in the appendix. When the filters are normalized and there is no bias, they are proportional to $M \|f\|_\infty$. In particular, they do not depend on the dimension $d_x$.

The proof use standard algebraic manipulations, along with two concentration inequalities. The first one exploits Dudley's inequality [44] to show that, for a fixed function $f$ and in the absence of random edges, $\mathcal{L}_{W,P} f$ is well approximated by its discrete counterpart. Note here that we do not seek a *uniform* proof with respect to a functional space, since the c-GCN is fixed. This allows us to obtain non-asymptotic rate while relaxing usual smoothness hypotheses [41]. This first concentration bound leads to the standard rate in $\mathcal{O}(1/\sqrt{n})$.

The second bound uses a fairly involved recent concentration inequality for normalized Laplacians of relatively sparse graphs with random edges derived in [25], which gives the term in $\mathcal{O}(1/\sqrt{\alpha_n n})$. Although this second term has a strictly worse convergence rate except in the dense case $\alpha_n \sim 1$, its multiplicative constant is strictly better, in particular it does not depend on the Minkowski dimension $d_x$. The condition $n \geqslant 1/\rho$, which suggests a polynomial concentration instead of the more traditional exponential one, comes from this part of the proof.

It is known in the literature that using the normalized Laplacian is often more appropriate than the adjacency matrix. If we where to use the latter, a normalization by $(\alpha_n n)^{-1}$ would be necessary [30]. However, $\alpha_n$ is rarely known, and can change from one case to the other. The normalized Laplacian is adaptive to $\alpha_n$ and does not require any normalization.

**Example of applications.** Invariant GCNs are typically used for regression or classification at the graph level. Theorem 1 shows that the output of a discrete GCN directly approaches that of the corresponding c-GCN. Equivariant GCNs are typically used for regression at the node level. Consider an ideal function $f^* : \mathcal{X} \to \mathbb{R}^{d_{out}}$ that is well approximated by an equivariant c-GCN $\Phi_{W,P}(f)$ in terms of $L^2(P)$-norm. Then, the error between the output of the discrete GCN $\Phi_A(Z)$ and the sampling of $f^*$ satisfies with high probability $\mathrm{MSE}_X (\Phi_A(Z), f^*) \leqslant \|\Phi_{W,P}(f) - f^*\|_{L^2(P)} + R_n + \mathcal{O}(n^{-\frac{1}{4}})$ using a triangle inequality, Theorem 1 and Hoeffding's inequality.

**Noisy or absent signal.** Until now, we have considered that the function $f$ was observed without noise. Noise can be handled by considering the Lipschitz properties of the GCN. For instance, in the invariant case, by Lemma **??** in Appendix **??**, we have $\left\| \bar{\Phi}_A(Z_1) - \bar{\Phi}_A(Z_2) \right\| \lesssim \frac{1}{\sqrt{n}} \|Z_1 - Z_2\|_F$. Hence, if the input signal is the noisy $z_i = f(x_i) + \nu_i$, where $\nu$ is centered iid noise, a GCN deviates from the corresponding c-GCN by an additional $n^{-1/2} \|(\nu_i)_i\|_F$, which converges to the standard deviation of the noise. Interestingly, the noise can be filtered out: for instance, if one inputs $\bar{Z} = LZ$ into the GCN, then by a concentration inequality it is not difficult to see that the smoothed noise term converges to 0, and the GCN converges to the c-GCN with smoothed input function $\bar{f} = \mathcal{L}_{W,P} f$.

In some cases such as spectral clustering [11], one does not have an input signal over the nodes, but has only access to the structure of the graph. In this case, several heuristics have been used in the literature, but a definitive answer is yet to emerge. For instance, a classical strategy is to use the (normalized) degrees of the graph $Z = A1_n/(\alpha_n n)$ as input signal [9, 11] (assuming for simplicity that $\alpha_n$ is known or estimated). In this case, using our proofs (Lemma **??** in the appendix) and the spectral concentration in [30], it is not difficult to show that a discrete GCN will converge to its countinuous version with the degree function $f = d_{W,P}$ as input. We will see in Prop. 3 in the next section that this leads to desirable stability properties.

## 4   Stability of GCNs to model deformations

Stability to deformations is an essential feature for the generalization properties of deep architectures. Mallat [34] studied the stability to small deformations of the wavelet-based scattering transform, which was extended to more generic learned convolutional network, *e.g.,* [5, 40], and tries to establish bounds of the following form for a signal representation $\Phi(\cdot)$:

$$\|\Phi(f_\tau) - \Phi(f)\| \lesssim N(\tau)\|f\|, \tag{10}$$

where $f_\tau(x) = f(x - \tau(x))$ is the deformed signal and $N(\tau)$ quantifies the size of the deformation, typically through norms of its jacobian $\nabla\tau$, such as $\|\nabla\tau\|_\infty = \sup_x \|\nabla\tau(x)\|$. As we have seen in the introduction, it is not clear how to extend the notion of deformation on discrete graphs [17, 19]. We show here that it can be done in the continuous world. We first derive generic stability bounds for discrete random graphs involving a Wasserstein-type metric between the corresponding c-GCNs, then derive bounds of the form (10) for c-GCNs by studying various notions of deformations of random graph models. We note that "spatial" deformations $x \mapsto x - \tau(x)$ are or course not the only possible choice for these models, and leave other types of perturbations to future work.

**From discrete to continuous stability.**   We first exploit the previous convergence result to deport the stability analysis from discrete to continuous GCNs. Let $G_1$ and $G_2$ be two random graphs with $n$ nodes drawn from models $\Gamma_1$ and $\Gamma_2$, and a GCN $\Phi$. In the invariant case, we can directly apply Theorem 1 and the triangle inequality to obtain that $\|\bar{\Phi}_{A_1}(Z_1) - \bar{\Phi}_{A_2}(Z_2)\| \leqslant \|\bar{\Phi}_{W_1,P_1}(f_1) - \bar{\Phi}_{W_2,P_2}(f_2)\| + 2R_n$, and study the robustness of $\bar{\Phi}_{W,P}(f)$ to deformations of the model. The equivariant case is more complex. A major difficulty, compared for instance to [31], is that, since we consider two different samplings $X_1$ and $X_2$, there are no implicit ordering over the nodes of $G_1$ and $G_2$, and one cannot directly compare the output signals of the equivariant GCN *e.g.,* in Frobenius norm. To compare two graph representations, a standard approach in the study of stability (and graph theory in general) has been to define a metric that minimizes over permutations $\sigma$ of the nodes (*e.g.,* [17, 19]), thus we define $\text{MSE}_\Sigma(Z, Z') \overset{\text{def.}}{=} \min_{\sigma \in \Sigma_n}(n^{-1}\sum_i \|Z_i - Z'_{\sigma(i)}\|^2)^{1/2}$. Theorem 2 relates this to a Wasserstein metric between the c-GCNs (proof in Appendix **??**).

**Theorem 2** (Finite-sample stability in the equivariant case). *Adopt the notations of Theorem 1. For $r = 1, 2$, define the distribution $Q_r = \Phi_{W_r,P_r}(f_r)_\sharp P_r$. With probability $1 - \rho$, we have*

$$\text{MSE}_\Sigma\left(\Phi_{A_1}(Z_1), \Phi_{A_2}(Z_2)\right) \leqslant \mathcal{W}_2(Q_1, Q_2) + R_n + C_1\left(n^{-\frac{1}{d_z}} + \left(C_2 + \sqrt[4]{\log\frac{1}{\rho}}\right)n^{-\frac{1}{4}}\right) \tag{11}$$

*where $C_1$ and $C_2$ are defined in the appendix. When $f_1$ and $f_2$ are piecewise Lipschitz, the last terms can be replaced by $C'_1(n^{-1/d_x} + (C'_2 + \sqrt[4]{\log(1/\rho)})n^{-1/4})$ for some $C'_1, C'_2$.*

In other words, we express stability in terms of a Wasserstein metric between the push-forwards of the measures $P_r$ by their respective c-GCNs. By definition, the l.h.s. of (11) is invariant to permutation of the graphs $G_r$. Moreover, for $\phi \in \Sigma_P$ by Prop. 2 we have $\Phi_{\phi\cdot W,P}(\phi \cdot f)_\sharp P = \Phi_{W,P}(f)_\sharp(\phi_\sharp P) = \Phi_{W,P}(f)_\sharp P$, and therefore the r.h.s. of (11) is also invariant to continuous permutation $\phi$.

We recover the rate $R_n$ from Theorem 1, as well as a term in $1/n^{1/4}$ and a term that depends on the dimension. In the relatively sparse case, the term in $1/\sqrt{\alpha_n n}$ in $R_n$ still has the slowest convergence rate. The proof uses classic manipulations in Optimal Transport [39], as well as concentration results of empirical distributions in Wasserstein norm [46]. In particular, it is known that the latter yields slow convergence rates with the dimension $n^{-1/d}$. While the $Q_r$'s live in $\mathbb{R}^{d_z}$, when the c-GCNs are Lipschitz we can replace $d_z$ by the Minkowski dimension of $\mathcal{X}$, which may be advantageous when $\mathcal{X}$ is a low-dimensional manifold.

In the rest of this section, we analyze the stability of c-GCNs to deformation of random graph models, directly through the Wasserstein bound above (or simple Euclidean norm in the invariant case). Finite-sample bounds are then obtained with Theorem 1 and 2.

**Stability of continuous GCNs: assumptions.** Assume from now on that $\mathcal{X} \subset \mathbb{R}^d$ equipped with the Euclidean norm. Given a diffeomorphism $\tau : \mathcal{X} \to \mathcal{X}$, we consider spatial deformations of random graph models of the form $(\mathrm{Id} - \tau)$, and aim at obtaining bounds of the form (10) for c-GCNs. Given a reference random graph model $\Gamma = (P, W, f)$, we may consider perturbations to $P$, $W$, or $f$, and thus define $W_\tau \stackrel{\text{def.}}{=} (\mathrm{Id} - \tau) \cdot W$, $P_\tau \stackrel{\text{def.}}{=} (\mathrm{Id} - \tau)_\sharp P$ and $f_\tau \stackrel{\text{def.}}{=} (\mathrm{Id} - \tau) \cdot f$. Of course, after deformation, we still consider that the assumptions on our random graph models (6) are verified. As can be expected, translation-invariant kernels $W$ such as Gaussian kernels or $\varepsilon$-graph kernels are particularly adapted to such deformations, therefore we will often make the following assumption:

$$W(x, x') = w(x - x'), \quad C_{\nabla w} \stackrel{\text{def.}}{=} \sup_{x \in \mathcal{X}} \int \left\| \nabla w \left( \tfrac{x - x'}{2} \right) \right\| \cdot \|x' - x\| \, dP(x') < \infty. \tag{A1}$$

We also define $C_W \stackrel{\text{def.}}{=} \sup_x \int |W(x, x')| dP(x') \leqslant c_{\max}$. While $C_W$ and $C_{\nabla w}$ are easily bounded when $W, \nabla w$ are bounded, they are typically much smaller than such naive bounds when $W$ and $\nabla w$ are well localized in space with fast decays, *e.g.,* for the Gaussian kernel or a smooth $\varepsilon$-graph kernel with compact support (for instance, in the latter case, $C_W$ is proportional to $\varepsilon c_{\max}$ instead of $c_{\max}$).

In the case where $P$ is replaced by $P_\tau$, some of our results will be valid beyond translation-invariant kernels. We will instead assume that $P_\tau$ has a density with respect to $P$, close to one: for all $x$,

$$q_\tau(x) \stackrel{\text{def.}}{=} \tfrac{dP_\tau}{dP}(x), \quad q_\tau(x), q_\tau(x)^{-1} \leqslant C_{P_\tau} < \infty, \quad N_P(\tau) \stackrel{\text{def.}}{=} \|q_\tau - 1\|_\infty. \tag{A2}$$

When $(\mathrm{Id} - \tau) \in \Sigma_P$, then we have $N_P(\tau) = 0$, so that $N_P(\tau)$ measures how much it deviates from such neutral elements and quantifies the size of deformations. In particular, when $P$ is proportional to the Lebesgue measure and $\|\nabla \tau\|_\infty < 1$, we have $q_\tau(x) = \det(I - \nabla\tau(x))^{-1}$; then, for small enough $\|\nabla\tau\|_\infty$, we obtain $N_P(\tau) \lesssim d\|\nabla\tau\|_\infty$, recovering the more standard quantity of Mallat [34]. In this case, we also have the bound $C_{P_\tau} \leqslant 2^d$ if we assume $\|\nabla\tau\|_\infty \leqslant 1/2$.

In the rest of the section, we will assume for simplicity that the considered GCNs $\Phi$ have zero bias at each layer. Unless otherwise written, $\|f\|$ refers to $L^2(P)$-norm. All the proofs are in Appendix **??**.

**Deformation of translation-invariant kernels.** We first consider applying deformations to the kernel $W$, which amounts to a perturbation to the edge structure of the graph. For GCNs, this affects the Laplacian operator used for the filters, and could be seen as a perturbation of the "graph shift operator" in the framework of Gama et al. [19]. The following result shows that in this case the stability of GCN representations, both invariant and equivariant, is controlled by $\|\nabla\tau\|_\infty$.

**Theorem 3** (Kernel deformation). *Consider a GCN representation $\Phi$ with no bias and a random graph $\Gamma = (P, W, f)$. Define $Q = \Phi_{W,P}(f)_\sharp P$ and $Q_\tau = \Phi_{W_\tau,P}(f)_\sharp P$. Assume (A1) and $\|\nabla\tau\|_\infty \leqslant 1/2$. We have*

$$\left. \begin{array}{c} \|\bar{\Phi}_{W_\tau,P}(f) - \bar{\Phi}_{W,P}(f)\| \\ \mathcal{W}_2(Q, Q_\tau) \end{array} \right\} \leqslant C(C_W + C_{\nabla w})\|f\|\|\nabla\tau\|_\infty, \tag{12}$$

*where $C$ is given in the appendix.*

**Deformation of the distribution.** Let us now consider perturbations of $P$ to $P_\tau$, which corresponds to a change in the node distribution. In practice, this may correspond to several, fairly different, "practical" situations. We describe two different frameworks below.

In shape analysis, $P$ may be supported on a manifold, and $P_\tau$ can then represent a deformation of this manifold, *e.g.,* a character that rigidly moves a body part. In this case in particular, we can expect $\|\tau\|_\infty$ to be large, but $\|\nabla\tau\|_\infty$ to be small (*i.e.,* large translation but small deformation). Moreover, if the kernel is translation-invariant, there will be little change in the structure of the generated graph. If additionally the input signal of the c-GCN is approximately deformed along with $P$, then one can expect the outputs to be stable, which we prove in the following theorem.

**Theorem 4** (Distribution deformation, translation-invariant case). *Consider a GCN representation $\Phi$ with no bias and a random graph $\Gamma = (P, W, f)$, along with a function $f'$. Define $Q = \Phi_{W,P}(f)_\sharp P$*

*and* $Q_\tau = \Phi_{W, P_\tau}(f')_\sharp P_\tau$. *Assume* (A1) *and* $\|\nabla \tau\|_\infty \leqslant 1/2$. *We have*

$$\left.\begin{array}{c} \left\|\bar\Phi_{W,P}(f) - \bar\Phi_{W,P_\tau}(f')\right\| \\ \mathcal{W}_2(Q, Q_\tau) \end{array}\right\} \leqslant C(C_W + C_{\nabla w})\|f\|\|\nabla\tau\|_\infty + C'\left\|f'_\tau - f\right\|, \tag{13}$$

*where* $C, C'$ *are given in the appendix.*

When $f = f'$ are both constant, or when $f' = (\text{Id} - \tau)^{-1} \cdot f$, that is, $f'$ is the mapping of the original signal $f$ on the deformed structure, then we have $\|f'_\tau - f\| = 0$. As mentioned before, in the absence of input signal, a standard choice is to take the degree functions as inputs [9, 11]. The next result shows that this choice also leads to the desired stability.

**Proposition 3.** *Assume* (A1) *and* $\|\nabla\tau\|_\infty \leqslant 1/2$. *If* $f = d_{W,P}$ *and* $f' = d_{W,P_\tau}$, *then we have* $\|f'_\tau - f\| \leqslant C_{\nabla w} \|\nabla\tau\|_\infty$.

Let us now take a look at the case where $W$ is not translation-invariant. We will then assume that $P_\tau$ has a density with respect to $P$, and in particular that it has the same support: one may for instance imagine a social network with a slightly changing distribution of user preferences, SBMs with changing community sizes, geometric random graphs [38], or graphons [32]. The analysis here being slightly more complex, we focus on invariant c-GCNs.

**Theorem 5** (Distribution deformation, non-translation-invariant case). *Consider a GCN representation* $\Phi$ *with no bias and a random graph* $\Gamma = (P, W, f)$. *Assume* (A2). *We have*

$$\left\|\bar\Phi_{W,P}(f) - \bar\Phi_{W,P_\tau}(f)\right\| \leqslant \left(CC_{P_\tau}^3 C_W + C'\right)\|f\|N_P(\tau), \tag{14}$$

*where* $C, C'$ *are given in the appendix.*

As mentioned above, in the case where $P$ is the Lebesgue measure, *e.g.*, for graphons [32], then we recover the quantity $N_P(\tau) \lesssim d\|\nabla\tau\|_\infty$.

**Deformations of the signal.** Finally, we consider deformations of the signal on the graph and show a bound similar to the ones in the Euclidean case (10). As can be seen in the proofs, this case is in fact a combination of the previous results (12) and (14); hence we must assume both (A1) and (A2) and obtain a dependence on both $\|\nabla\tau\|_\infty$ and $N_P(\tau)$. Once again we focus on invariant c-GCNs with pooling, similar to classical scattering transform [34].

**Proposition 4** (Signal deformation). *Consider a GCN representation* $\Phi$ *with no bias and a random graph* $\Gamma = (P, W, f)$. *Assume* (A1), (A2), *and* $\|\nabla\tau\|_\infty \leqslant 1/2$. *We have*

$$\left\|\bar\Phi_{W,P}(f) - \bar\Phi_{W,P}(f_\tau)\right\| \leqslant (CC_{P_\tau}^{1/2}(C_W + C_{\nabla w})\|\nabla\tau\|_\infty + \left(CC_{P_\tau}^3 C_W + C'\right)N_P(\tau))\|f\|, \tag{15}$$

*where* $C, C'$ *are given in the appendix.*

When $P$ is proportional to the Lebesgue measure, since $N_P(\tau)$ is controlled by $\|\nabla\tau\|_\infty$, the GCN is invariant to translations and stable to deformations, similar to Euclidean domains [34]. We note that studies of stability are often balanced by discussions on how the representation preserves signal (*e.g.,* [34, 5, 18]). In our context, the empirical success of GCNs suggests that these representations maintain good discrimination and approximation properties, though a theoretical analysis of such properties for GCNs is missing and provides an important direction for future work.

## 5   Numerical experiments

In this section, we provide simple numerical experiments on synthetic data that illustrate the convergence and stability of GCNs. We consider untrained GCNs with random weights in order to assess how these properties result from the choice of architecture rather than learning. The code is accessible at https://github.com/nkeriven/random-graph-gnn.

**Convergence.** Fig. 1 shows the convergence of an equivariant GCN toward a continuous function on a $\varepsilon$-graph with nodes sampled on a 3-dimensional manifold. We take a constant input signal $f = 1$ here to only assess the effect of the manifold shape. We then examine the effect of the sparsity level on convergence on the corresponding invariant GCN, taking an average of several experiments with high $n$ and $\alpha_n = 1$ as an approximation of the "true" unknown limit value. As expected, the convergence is slower for sparse graphs, however we indeed observe convergence to the *same* output for all values of $\alpha_n$.

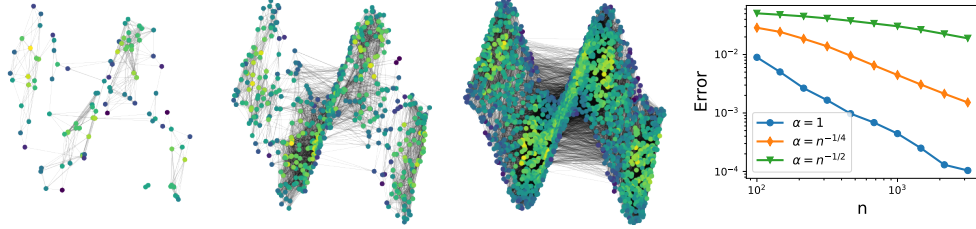

Figure 1: Illustration of convergence of a GCN on random graphs with 3D latent positions and input signal $f = 1$. Left: output signal with growing number of nodes. Right: convergence with different sparsity levels $\alpha_n$.

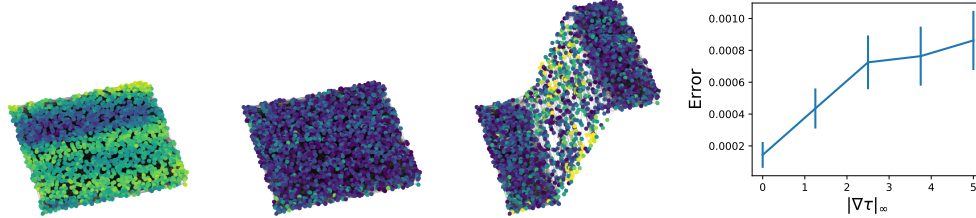

Figure 2: Illustration of stability of a GCN on random graphs with 3D latent positions. From left to right: output signal; difference with the output on the same latent positions but a new drawing of the random edges; difference with the output on deterministically deformed latent positions and corresponding drawing of random edges; difference in output signal of an invariant GCN with respect to the amplitude of the deformation, averaged over 20 experiments.

**Stability.** In Fig. 2, we illustrate the stability of GCNs to deformations. We first examine the variations in the output of an equivariant GCN when only re-drawing the random edges, with or without modifying the latent positions (in a deterministic manner). We indeed observe that regions that are only translated, such as the "flat" parts of the surface, yield stable output, while deformed regions lead to a deformed output signal. We then verify that a larger deformation leads to a larger distance in output.

## 6 Conclusion and outlooks

GCNs have proved to be efficient in identifying large-scale structures in graphs and generalizing across graphs of different sizes, which can only partially be explained with discrete graph notions like isomorphisms [48] or stability with permutation-minimizing metrics [18]. In contrast, we have shown that combining them with random models of large graphs allows us to define intuitive notions of deformations and stability in the continuous world like the Euclidean case [34, 5, 40], with direct applications in community-based social networks or shape analysis on point clouds. For this we derived non-asymptotic convergence bounds, valid on relatively sparse random graphs with non-smooth kernels, and new tools like a Wasserstein-type stability bounds on equivariant c-GCNs.

We believe our work to be a first step toward a better understanding of GNNs on large graphs, with many potential outlooks. First, it would be useful to improve the dependence of our bounds on regularity properties of the filters, as done in [19] for the discrete setting, while preserving the mild dependence on the number of filters. In the same vein, finer results may be obtained in particular cases: *e.g.,* the case where $\mathcal{X}$ is a sub-manifold can be studied under the light of Riemannian geometry, stability bounds on SBMs may be expressed with a direct dependence on their parameters, or more explicit stability bounds may be obtained when the (c-)GCN is a structured architecture like the scattering transform on graphs [17]. Convergence results can also be obtained for many other models of random graphs like $k$-Nearest Neighbor graphs [10]. Finally, while we focus on stability in this paper, as mentioned above the *approximation power* of GCNs (beyond untractable universality [24]) can also be expressed through that of their continuous counterpart, and characterizing which functions are computable by a c-GCN (*e.g.,* with growing width or number of layers) is of foremost importance.

## Broader Impact

Graph Neural Networks have been used to many applications, including computer vision, generative models in NLP or protein prediction to cite only a few. Thus, our work is included in a wide literature whose societal impact and ethical considerations are not one-sided. We provide here a theoretical understanding of the behaviour of large random graphs, with the most natural application is community detection in social science [3]. Our contributions, of a theoretical nature, are far from a direct impact in our opinion. We do not see continuous GCNs applied directly in the forseeable future otherwise as a proxy for the study of classic GCNs. Nevertheless, as a stability result, it is a step forward to handle adversarial attacks as highlighted in [19].

## Acknowledgements

AB acknowledges support from the European Research Council (grant SEQUOIA 724063). SV is partly supported by ANR JCJC GraVa (ANR-18-CE40-0005).

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
