[Supplementary Material]

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

## Supplementary material

In Appendix A, we introduce additional notations and objects that will be used in the proofs. In Appendix B, we study the equivariance of GCNs and prove Props 1 and 2. In Appendix C, we prove Theorem 1 on the convergence of GCNs. In Appendix D, we prove the Wasserstein bound in Theorem 2. In Appendix E, we derive the stability bounds of Section 4. Finally, in Appendix F, we give technical concentration bounds and in Appendix G we provide some third-party results for completeness.

## A   Notations

Given a GCN, we define some bounds on its parameters that will be used in the multiplicative constants of the theorem. Recall that the filters are written $h_{ij}^{(\ell)}(\lambda) = \sum_{k=0}^{\infty} \beta_{ijk}^{(\ell)} \lambda^k$. We define $B_k^{(\ell)} = \left(\beta_{ijk}^{(\ell)}\right)_{ji} \in \mathbb{R}^{d_{\ell+1} \times d_\ell}$ the matrix containing the order-$k$ coefficients, and by $B_{k,|\cdot|}^{(\ell)} = \left(\left|\beta_{ijk}^{(\ell)}\right|\right)_{ji}$ the same matrix with absolute value on all coefficients. Then, we define the following bounds:

$$H_2^{(\ell)} = \sum_k \left\| B_k^{(\ell)} \right\| \qquad\qquad H_{\partial,2}^{(\ell)} = \sum_k \left\| B_k^{(\ell)} \right\| k$$

$$H_\infty^{(\ell)} = \sum_k \left\| B_{k,|\cdot|}^{(\ell)} \right\| \left(\frac{2c_{\max}}{c_{\min}}\right)^k \qquad\qquad H_{\partial,\infty}^{(\ell)} = \sum_k \left\| B_k^{(\ell)} \right\| k \left(\frac{2c_{\max}}{c_{\min}}\right)^{k-1}$$

which all converge by our assumptions on the $\beta_k$. We may also denote $H_2$ by $H_{L^2(P)}$ for convenience but this quantity does not depend on $P$. Note that, only for $H_\infty$, we use the spectral norm of the matrix $B_{k,|\cdot|}$ with non-negative coefficients, which is suboptimal compared to using $B_k$. This is due to a part of our analysis where we do not operate in a Hilbert space but only in a Banach space $\mathcal{B}(\mathcal{X})$, see Lemma 6. We also define $\left\| b^{(\ell)} \right\| = \sqrt{\sum_j (b_j^{(\ell)})^2}$.

Given $X$, we define the empirical degree function

$$d_X = d_{W,X} \stackrel{\text{def.}}{=} \frac{1}{n} \sum_i W(\cdot, x_i) \qquad\qquad (16)$$

Which will be denoted by $d_X$ when the kernel is clear. Although $d_{W,P}$ is bounded away from 0 by the assumption (6), this is not necessarily the case for $d_{W,X}$. This is however true with high probability, as shown by the following Lemma.

**Lemma 1.** *Let $\Gamma$ be a model of random graphs. There is a universal constant $C$ such that, if*

$$n \geqslant C D_{\mathcal{X}}(\rho)^2 \qquad\qquad (17)$$

*where $D_{\mathcal{X}}(\rho) = \frac{c_{\text{Lip.}}}{c_{\min}} \sqrt{d_x} + \frac{c_{\max} + c_{\text{Lip.}}}{c_{\min}} \sqrt{\log \frac{n_{\mathcal{X}}}{\rho}}$, then with probability $1 - \rho$, $d_{W,X} \geqslant c_{\min}/2 > 0$.*

*Proof.* Apply Lemma 4 with $f = 1$ to obtain the result. $\qquad\qquad\square$

For $W$ and $X$ such that $d_{W,X} > 0$, we define the following empirical Laplacian operator:

$$\mathcal{L}_X f = \mathcal{L}_{W,X} f \stackrel{\text{def.}}{=} \frac{1}{n} \sum_i \frac{W(\cdot, x_i)}{\sqrt{d_X(\cdot) d_X(x_i)}} f(x_i) \qquad\qquad (18)$$

which we will also denote by $\mathcal{L}_X$ when $W$ is clear. Assuming that $d_X \geqslant c_{\min}/2$, $\mathcal{L}_{W,X}$ is a bounded operator and $\|\mathcal{L}_{W,X}\|_\infty \leqslant \frac{2c_{\max}}{c_{\min}}$.

Given $X = \{x_1, \ldots, x_n\}$ and any dimension $d$, we denote by $S_X$ the normalized sampling operator acting on functions $f : \mathcal{X} \to \mathbb{R}^d$ defined by $S_X f \stackrel{\text{def.}}{=} \frac{1}{\sqrt{n}}[f(x_1), \ldots, f(x_n)] \in \mathbb{R}^{n \times d}$. The normalizing factor $\frac{1}{\sqrt{n}}$ is natural: we have $\|S_X f\|_F \leqslant \|f\|_\infty$ and by the Law of Large Numbers $\|S_X f\|_F \to \|f\|_{L^2(P)}$ a.s. Finally, given $X$ and $W$, we define $W(X) \stackrel{\text{def.}}{=} (W(x_i, x_j))_{ij} \in \mathbb{R}^{n \times n}$, and remark that $L(W(X)) \circ S_X = S_X \circ \mathcal{L}_{W,X}$.

## B  Invariance and equivariance

*Proof of Prop. 1.* The proof is immediate, by observing that $L(\sigma \cdot A) = \sigma \cdot L(A)$, therefore $h(L(\sigma \cdot A))(\sigma \cdot Z) = \sigma \cdot (h(L(A))Z)$, and permutations commute with the pointwise activation function. For the invariant case, we just observe the final pooling on the equivariant case. $\qquad\square$

*Proof of Prop. 2.* Let us first observe that the degree function is such that

$$d_{W,P}(\phi(x)) = \int W(\phi(x), x')dP(x') = \int (\phi \cdot W)(x, x')d(\phi^{-1})_\sharp P(x') = d_{\phi \cdot W,(\phi^{-1})_\sharp P}(x)$$

Then, we have

$$\begin{aligned}
\phi \cdot (\mathcal{L}_{W,P}f)(x) &= \int \frac{W(\phi(x), x')}{\sqrt{d_{W,P}(\phi(x))d_{W,P}(x')}}f(x')dP(x') \\
&= \int \frac{(\phi \cdot W)(x, x')}{\sqrt{d_{\phi \cdot W,(\phi^{-1})_\sharp P}(x)d_{\phi \cdot W,(\phi^{-1})_\sharp P}(x')}}(\phi \cdot f)(x')d(\phi^{-1})_\sharp P(x') \\
&= \mathcal{L}_{\phi \cdot W,(\phi^{-1})_\sharp P}(\phi \cdot f)
\end{aligned}$$

Then, by recursion, we have $\mathcal{L}^k_{\phi \cdot W,(\phi^{-1})_\sharp P}(\phi \cdot f) = \mathcal{L}^{k-1}_{\phi \cdot W,(\phi^{-1})_\sharp P}(\phi \cdot (\mathcal{L}_{W,P}f)) = \ldots = \phi \cdot \mathcal{L}^k_{W,P}f$, and the same is true for filters $h(\mathcal{L})$. We conclude by observing that permutation commutes with pointwise non-linearity: $\rho \circ (\phi \cdot f) = \phi \cdot (\rho \circ f) = \rho \circ f \circ \phi$. The invariant case follows with a final integration against $P$. $\qquad\square$

## C  Convergence of GCNs: proof of Theorem 1

We are going to prove Theorem 1 with the following constants:

$$\begin{aligned}
C_1 &\propto \frac{c_{\max} + c_{\text{Lip.}}}{c_{\min}} \sum_{\ell=0}^{M-1} C^{(\ell)} H^{(\ell)}_{\partial,\infty} \prod_{s=\ell+1}^{M-1} H^{(s)}_2, \\
C_2 &\propto \frac{c_{\max}}{c^2_{\min}} \sum_{\ell=0}^{M-1} C^{(\ell)} H^{(\ell)}_{\partial,2} \prod_{s=\ell+1}^{M-1} H^{(s)}_2, \\
C_3 &\propto C^{(M)}
\end{aligned}$$

$$\text{with} \quad C^{(\ell)} \stackrel{\text{def.}}{=} \|\theta\| \left( \|f\|_\infty \prod_{s=0}^{\ell-1} H^{(s)}_\infty + \sum_{s=0}^{\ell-1} \left\| b^{(s)} \right\| \prod_{p=s+1}^{\ell-1} H^{(p)}_\infty \right) \tag{19}$$

The proof will mainly rely on an application of Dudley's inequality [44, Thm 8.1.6] (Lemma 5 in Appendix F) and a recent spectral concentration inequality for normalized Laplacian in relatively sparse graphs (Theorem 6 in Appendix G).

*Proof.* We begin the proof by the equivariant case, the invariant case will simply use an additional concentration inequality. Denoting by $Z^{(\ell)}$ (resp. $f^{(\ell)}$) the signal at each layer of the GCN (resp. the function at each layer of the c-GCN), we have

$$\text{MSE}_X \left( \Phi_A(Z), \Phi_{W,P}(f) \right) = \left\| \frac{\Phi_A(Z)}{\sqrt{n}} - S_X \Phi_{W,P}(f) \right\|_F \leqslant \|\theta\| \left\| \frac{Z^{(M)}}{\sqrt{n}} - S_X f^{(M)} \right\|_F$$

where we recall that $S_X$ is the normalized sampling operator (see App. A). We therefore seek to bound that last term.

Assume that the following holds with probability $1 - \rho$: for all $0 \leqslant \ell \leqslant M - 1$

$$\sqrt{\sum_j \left\| \sum_i \left( h^{(\ell)}_{ij}(L) S_X f^{(\ell)}_i - S_X h^{(\ell)}_{ij}(\mathcal{L}_{W,P}) f^{(\ell)}_i \right) \right\|^2} \leqslant \Delta^{(\ell)} \tag{20}$$

Then, using (4), Lemma 6, and the fact that for unnormalized sampling $\sqrt{n}S_X \circ \rho = \rho \circ (\sqrt{n}S_X)$, we can show by recursion that $\left\| \frac{Z^{(\ell)}}{\sqrt{n}} - S_X f^{(\ell)} \right\|_F \leqslant \varepsilon_\ell$ implies

$$
\left\| \frac{Z^{(\ell+1)}}{\sqrt{n}} - S_X f^{(\ell+1)} \right\|_F
$$

$$
= \left( \sum_j \left\| \frac{1}{\sqrt{n}} \rho \Big( \sum_{i=1}^{d_\ell} h_{ij}^{(\ell)}(L) z_i^{(\ell)} + b_j^{(\ell)} 1_n \Big) - S_X \rho \Big( \sum_{i=1}^{d_\ell} h_{ij}^{(\ell)}(\mathcal{L}_{W,P}) f_i^{(\ell)} + b_j^{(\ell)} 1(\cdot) \Big) \right\|^2 \right)^{\frac{1}{2}}
$$

$$
= \left( \sum_j \frac{1}{n} \left\| \rho \Big( \sum_{i=1}^{d_\ell} h_{ij}^{(\ell)}(L) z_i^{(\ell)} + b_j^{(\ell)} 1_n \Big) - \rho \Big( \sqrt{n} S_X \Big( \sum_{i=1}^{d_\ell} h_{ij}^{(\ell)}(\mathcal{L}_{W,P}) f_i^{(\ell)} + b_j^{(\ell)} 1(\cdot) \Big) \Big) \right\|^2 \right)^{\frac{1}{2}}
$$

$$
\leqslant \left( \sum_j \left\| \sum_{i=1}^{d_\ell} h_{ij}^{(\ell)}(L) \frac{z_i^{(\ell)}}{\sqrt{n}} - S_X h_{ij}^{(\ell)}(\mathcal{L}_{W,P}) f_i^{(\ell)} \right\|^2 \right)^{\frac{1}{2}}
$$

$$
\leqslant \left( \sum_j \left\| \sum_{i=1}^{d_\ell} h_{ij}^{(\ell)}(L) \left( \frac{z_i^{(\ell)}}{\sqrt{n}} - S_X f_i^{(\ell)} \right) \right\|^2 \right)^{\frac{1}{2}}
$$

$$
+ \left( \sum_j \left\| \sum_{i=1}^{d_\ell} h_{ij}^{(\ell)}(L) S_X f_i^{(\ell)} - S_X h_{ij}^{(\ell)}(\mathcal{L}_{W,P}) f_i^{(\ell)} \right\|^2 \right)^{\frac{1}{2}}
$$

$$
\leqslant \varepsilon_{\ell+1} \overset{\text{def.}}{=} H_2^{(\ell)} \varepsilon_\ell + \Delta^{(\ell)}
$$

Since $\frac{Z^{(0)}}{\sqrt{n}} = S_X f^{(0)}$ we have $\varepsilon_0 = 0$ and an easy recursion shows that

$$
\left\| \frac{Z^{(M)}}{\sqrt{n}} - S_X f^{(M)} \right\|_F \leqslant \sum_{\ell=0}^{M-1} \Delta^{(\ell)} \prod_{s=\ell+1}^{M-1} H_2^{(s)} \tag{21}
$$

We now need to prove that (20) holds with probability $1 - \rho$ for all $\ell$ with the appropriate $\Delta^{(\ell)}$. Recall that $L(W(X)) \circ S_X = S_X \circ \mathcal{L}_{W,X}$, and that by (17), with probability $1 - \rho/2$ we have $\|\mathcal{L}_{W,X}\|_\infty \leqslant \frac{2c_{\max}}{c_{\min}}$. Assuming this is satisfied, by Lemma 6 we have

$$
\sqrt{ \sum_j \left\| \sum_i \left( h_{ij}^{(\ell)}(L) S_X f_i^{(\ell)} - S_X h_{ij}^{(\ell)}(\mathcal{L}_{W,P}) f_i^{(\ell)} \right) \right\|^2 }
$$

$$
\leqslant \sqrt{ \sum_j \left\| \sum_i \left( h_{ij}^{(\ell)}(L) - h_{ij}^{(\ell)}(L(W(X))) \right) S_X f_i^{(\ell)} \right\|^2 }
$$

$$
+ \sqrt{ \sum_j \left\| \sum_i S_X \left( h_{ij}^{(\ell)}(\mathcal{L}_{W,X}) - h_{ij}^{(\ell)}(\mathcal{L}_{W,P}) \right) f_i^{(\ell)} \right\|^2 }
$$

$$
\leqslant H_{\partial,2}^{(\ell)} \|L - L(W(X))\| \left\| f^{(\ell)} \right\|_\infty
$$

$$
+ \sum_k \|B_k\| \sqrt{ \sum_i \left( \sum_{\ell=0}^{k-1} \left( \frac{2c_{\max}}{c_{\min}} \right)^\ell \left\| (\mathcal{L}_{W,X} - \mathcal{L}_{W,P}) \mathcal{L}_{W,P}^{k-1-\ell} f_i^{(\ell)} \right\|_\infty \right)^2 } \tag{22}
$$

The first term in (22) is handled with a recent concentration inequality for normalized Laplacian in the relatively sparse graphs with random edges [25], recalled as Theorem 6 in Appendix G. We use the following version.

**Corollary 1** (of Theorem 6). *Assume* (17) *is satisfied and* $n \geqslant 1/\rho$ *for simplicity. There is a universal constant $C$ such that, if*

$$\alpha_n \geqslant \frac{Cc_{\max}}{c_{\min}^2} \cdot \frac{\log n}{n} \tag{23}$$

*Then, with probability at least $1 - \rho$, we have*

$$\|L - L(W(X))\| \lesssim \frac{c_{\max}}{c_{\min}^2} \cdot \frac{1}{\sqrt{\alpha_n n}}$$

*Proof.* By (17) with the appropriate constant, with probability $1 - \rho/2$ we have $d_X \geqslant c_{\min}/2$, we can therefore apply Theorem 6 to bound $\|L - L(W(X))\|$ conditionally on $X$ using $c \sim 1 + \frac{\log(1/\rho)}{\log(n)} \sim 1$, then use a union bound to conclude. $\qquad\square$

We now bound the second term in (22). Define $\rho_k = \frac{C\rho}{(k+1)^2 \sum_\ell d_\ell}$ with $C$ such that $\sum_{k\ell} d_\ell \rho_k = \rho/4$ (even when the filters are not of finite order). Using an application of Dudley's inequality detailed in Lemma 5 in Appendix F and a union bound, we obtain with probability $1 - \rho/4$ that: for all $i, \ell, k$, we have

$$\left\|(\mathcal{L}_{W,X} - \mathcal{L}_{W,P})\mathcal{L}_{W,P}^k f_i^{(\ell)}\right\|_\infty \lesssim \frac{c_{\max} \left\|\mathcal{L}_{W,P}^k f_i^\ell\right\|_\infty D_{\mathcal{X}}(\rho_k)}{c_{\min}\sqrt{n}}$$

$$\leqslant \left(\frac{c_{\max}}{c_{\min}}\right)^k \frac{c_{\max} \left\|f_i^\ell\right\|_\infty D_{\mathcal{X}}\left(\frac{C\rho}{(k+1)^2 \sum_\ell d_\ell}\right)}{c_{\min}\sqrt{n}}$$

$$\lesssim \left(\frac{2c_{\max}}{c_{\min}}\right)^k \frac{(c_{\max} + c_{\text{Lip.}})\left\|f_i^\ell\right\|_\infty D_{\mathcal{X}}\left(\frac{\rho}{\sum_\ell d_\ell}\right)}{c_{\min}\sqrt{n}}$$

Coming back to the second term of (22), with probability $1 - \rho/4$:

$$\sum_k \|B_k\| \sqrt{\sum_i \left(\sum_{\ell=0}^{k-1} \left(\frac{2c_{\max}}{c_{\min}}\right)^\ell \left\|(\mathcal{L}_{W,X} - \mathcal{L}_{W,P})\mathcal{L}_{W,P}^{k-1-\ell} f_i^{(\ell)}\right\|_\infty\right)^2}$$

$$\lesssim \frac{(c_{\max} + c_{\text{Lip.}})D_{\mathcal{X}}\left(\frac{\rho}{\sum_\ell d_\ell}\right)}{c_{\min}\sqrt{n}} \sum_k \|B_k\| k \left(\frac{2c_{\max}}{c_{\min}}\right)^k \sqrt{\sum_i \left\|f_i^{(\ell)}\right\|_\infty^2}$$

$$\leqslant \frac{(c_{\max} + c_{\text{Lip.}})D_{\mathcal{X}}\left(\frac{\rho}{\sum_\ell d_\ell}\right)}{c_{\min}\sqrt{n}} H_{\partial,\infty}^{(\ell)} \left\|f^{(\ell)}\right\|_\infty$$

At the end of the day we obtain that with probability $1 - \rho$, (20) is satisfied with

$$\Delta^{(\ell)} \propto \left\|f^{(\ell)}\right\|_\infty \left(\frac{H_{\partial,2}^{(\ell)} c_{\max}}{c_{\min}^2 \sqrt{\alpha_n n}} + \frac{H_{\partial,\infty}^{(\ell)}(c_{\max} + c_{\text{Lip.}})D_{\mathcal{X}}\left(\frac{\rho}{\sum_\ell d_\ell}\right)}{c_{\min}\sqrt{n}}\right)$$

We then use Lemma 8 to bound $\left\|f^{(\ell)}\right\|_\infty$ and conclude.

Finally, in the invariant case we have

$$\left\|\bar{\Phi}_A(Z) - \bar{\Phi}_{W,P}(f)\right\| \leqslant \text{MSE}_X\left(\Phi_A(Z), \Phi_{W,P}(f)\right) + \|\theta\| \left\|\frac{1}{n}\sum_i f^{(M)}(x_i) - \mathbb{E}f^{(M)}(X)\right\|$$

Using a vector Hoeffding's inequality (Lemma 11) and a bound on $\left\|f^{(M)}\right\|_\infty$ by Lemma 8 we bound the second term and conclude. $\qquad\square$

## D  Wasserstein convergence: proof of Theorem 2

We are going to prove Theorem 2 with the following constants:

$$C_1 \propto \|\theta\| \left( \max_{r=1,2} \|f_r\|_\infty \prod_{s=0}^{\ell-1} H_\infty^{(s)} + \sum_{s=0}^{\ell-1} \left\| b^{(s)} \right\| \prod_{p=s+1}^{\ell-1} H_\infty^{(p)} \right) \qquad C_2 = 27^{d_z/4} \qquad (24)$$

$$C_1' \propto (n_\mathcal{X} n_f)^{\frac{1}{d_x}} \max_{r=1,2} D_r \qquad\qquad\qquad C_2' = 27^{d_x/4}$$

where $D_r$ are defined as (36) with the function $f_r$ as input. The proof will mainly rely on results of the concentration rate of the empirical distribution of *iid* data to its true value in Wasserstein norm [46] (Theorem 7 in Appendix G).

*Proof.* For $r = 1, 2$, define $y_{r,i} = \Phi_{W_r, P_r}(f_r)(x_{r,i})$ which are drawn *iid* from $Q_r \overset{\text{def}}{=} \Phi_{W_r, P_r}(f_r)_\sharp P_r$, and we denote by $\hat{Q}_r = n^{-1} \sum_i \delta_{y_{r,i}}$ the empirical distributions. By Theorem 1 and triangle inequality, we have

$$\min_\sigma \sqrt{n^{-1} \sum_i \left\| \Phi_{A_1}(Z_1)_i - \Phi_{A_2}(Z_2)_{\sigma(i)} \right\|^2} \leqslant \min_\sigma \sqrt{\frac{1}{n} \sum_i \left\| y_{1,i} - y_{2,\sigma(i)} \right\|^2} + 2R_n$$

The first term is known, among other appellations, as the so-called "Monge" formulation of optimal transport (OT) [39, Chap. 2]. For uniform weights, as is the case here, it is known that Monge formulation of OT is equivalent to its Kantorovich relaxation, which in turns gives the traditional Wasserstein metric: by [39, Prop. 2.1], we have

$$\sqrt{\frac{1}{n} \sum_i \left\| y_{1,i} - y_{2,\sigma(i)} \right\|^2} = \mathcal{W}_2(\hat{Q}_1, \hat{Q}_2) \leqslant \mathcal{W}_2(Q_1, Q_2) + \sum_{r=1,2} \mathcal{W}_2(\hat{Q}_r, Q_r)$$

We must therefore bound the distance between an empirical distribution $\hat{Q}$ and its true value $Q = g_\sharp P$ for some function $g = \Phi_{W,P}(f)$.

The distribution $Q$ is supported on $g\mathcal{X} \subset \mathbb{R}^{d_z}$, which is bounded by $\|g\|_\infty$ which can be bounded by Lemma 8. Hence its covering numbers are as $N(g\mathcal{X}, \varepsilon, \|\cdot\|) \leqslant (\|g\|_\infty / \varepsilon)^{d_z}$. We can then conclude by Theorem 7.

When $f$ is $(c_f, n_f)$-piecewise Lipschitz however, by Lemma 9 $g$ is $(C, n_\mathcal{X} n_f)$-Lipschitz where $C$ is defined as (36), and in this case it is easy to see that the covering numbers of $g\mathcal{X}$ also satisfy $N(g\mathcal{X}, \varepsilon, \|\cdot\|) \leqslant n_\mathcal{X} n_f (C/\varepsilon)^{d_x}$. Applying again Theorem 7, we conclude. $\qquad\square$

## E  Stability

In this section, norms $\|\cdot\|$ always refer to $L^2(P)$ norms (for functions and operators), and we drop the subscript for simplicity. We denote the pooling operator by $U_P f = \int f dP$. We denote $U = U_P$ and $\mathcal{L} = \mathcal{L}_{W,P}$ for short. For $\tau : \mathcal{X} \to \mathcal{X}$, we denote $W_\tau = (\mathrm{Id} - \tau) \cdot W$, $P_\tau = (\mathrm{Id} - \tau)_\sharp P$, and $f_\tau = (\mathrm{Id} - \tau) \cdot f$. Then we define the shorthands $\mathcal{L}_{W_\tau} = \mathcal{L}_{W_\tau, P}$, $\mathcal{L}_{P_\tau} = \mathcal{L}_{W, P_\tau}$, the composition operator $T_\tau f = (\mathrm{Id} - \tau) \cdot f$ and $U_\tau = U_{P_\tau}$. Finally, we define $A$ and $A_\tau$ the integral operators of kernels $W$ and $W_\tau$ w.r.t. the measure $P$, the corresponding diagonal degree operators by $D$ and $D_{W_\tau}$, so that we have $\mathcal{L} = D^{-1/2} A D^{-1/2}$ and similarly for $\mathcal{L}_{W_\tau}$. Similarly, we define $D_{P_\tau}$ the diagonal degree operator of $W$ but with respect to $P_\tau$.

We observe that for two functions $f, f'$ and any $P$, we have

$$\left. \begin{array}{c} \left| \int f dP - \int f' dP \right| \\ \mathcal{W}_2(f_\sharp P, f'_\sharp P) \end{array} \right\} \leqslant \|f - f'\|_{L^2(P)} \qquad (25)$$

The second inequality is immediate by considering the definition $\mathcal{W}_2^2(f_\sharp P, f'_\sharp P) = \inf \left\{ \mathbb{E} \|f(X) - f'(Y)\|^2 \mid X \sim P, Y \sim P \right\}$ and taking $X = Y$ as a coupling. We will therefore manipulate mainly $L^2(P)$ norms.

### E.1 Proof of Theorem 3 (deformation change to $W$)

We are going to prove Theorem 3 with constant

$$C \propto \|\theta\| \, c_{\min}^{-2} \sum_{\ell=0}^{M-1} H_{\partial,2}^{(\ell)} \prod_{s=0, \; s \neq \ell}^{M-1} H_2^{(s)} \tag{26}$$

In this setting, we have two random graph models whose only difference is in the choice of kernel, from $W$ to $W_\tau$, while fixing $P$ and $f$. This in turn leads to a change in the Laplacian, which is the main quantity that we will need to control.

Since we have assumed the bias to be zero, we have the following Lemma, which we apply with $f = f'$ in this section.

**Lemma 2.** *We have*

$$\|\Phi_{W,P}(f) - \Phi_{W_\tau,P}(f')\| \leqslant C \, \|f\| \, \|\mathcal{L} - \mathcal{L}_{W_\tau}\| + C' \, \|f - f'\|$$

*with*

$$C = \|\theta\| \sum_{\ell=0}^{M-1} H_{\partial,2}^{(\ell)} \prod_{s=0, \; s \neq \ell}^{M-1} H_2^{(s)} \tag{27}$$

$$C' = \|\theta\| \prod_{\ell=0}^{M-1} H_2^{(\ell)} \tag{28}$$

*Proof.* Denoting $f^{(\ell)}$ and $(f^{(\ell)})'$ the functions at each layer, we have using Lemma 6 and (4):

$$\left\| f^{(\ell)} - (f^{(\ell)})' \right\| \leqslant \sqrt{ \sum_j \left\| \sum_{i=1}^{d_{\ell-1}} h_{ij}^{(\ell-1)}(\mathcal{L}) f_i^{(\ell-1)} - h_{ij}^{(\ell-1)}(\mathcal{L}_{W_\tau})(f_i^{(\ell-1)})' \right\|^2 }$$

$$\leqslant \sqrt{ \sum_j \left\| \sum_{i=1}^{d_{\ell-1}} (h_{ij}^{(\ell-1)}(\mathcal{L}) - h_{ij}^{(\ell-1)}(\mathcal{L}_{W_\tau})) f_i^{(\ell-1)} \right\|_*^2 }$$

$$+ \sqrt{ \sum_j \left\| \sum_{i=1}^{d_{\ell-1}} h_{ij}^{(\ell-1)}(\mathcal{L}_{W_\tau})(f_i^{(\ell-1)} - (f_i^{(\ell-1)})') \right\|^2 }$$

$$\leqslant H_{\partial,2}^{(\ell-1)} \|\mathcal{L} - \mathcal{L}_{W_\tau}\| \left\| f^{(\ell-1)} \right\| + H_2^{(\ell-1)} \left\| f^{(\ell-1)} - (f^{(\ell-1)})' \right\|$$

An easy recursion and Lemma 8 give the result. $\qquad\square$

The rest of the proof then consists in obtaining a bound on the quantity $\|\mathcal{L}_{W_\tau} - \mathcal{L}\|$ in $L^2(P)$. We have

$$\mathcal{L}_{W_\tau} - \mathcal{L} = D_{W_\tau}^{-\frac{1}{2}} A_\tau D_{W_\tau}^{-\frac{1}{2}} - D^{-\frac{1}{2}} A D^{-\frac{1}{2}}$$

$$= D_{W_\tau}^{-\frac{1}{2}} A_\tau (D_{W_\tau}^{-\frac{1}{2}} - D^{-\frac{1}{2}}) + D_{W_\tau}^{-\frac{1}{2}} (A_\tau - A) D^{-\frac{1}{2}} + (D_{W_\tau}^{-\frac{1}{2}} - D^{-\frac{1}{2}}) A D^{-\frac{1}{2}}. \tag{29}$$

We now bound different operators in this decomposition separately.

**Bound on $\|A_\tau - A\|$.** Define

$$k(x, x') = w(x - \tau(x) - x' + \tau(x')) - w(x - x'), \tag{30}$$

so that $A_\tau - A$ is an integral operator with kernel $k$. We have, by the fundamental theorem of calculus,

$$k(x, x') = \int_0^1 \langle \nabla w(x - x' + t(\tau(x') - \tau(x))), \tau(x') - \tau(x) \rangle dt.$$

Now, note that we have

$$|\tau(x') - \tau(x)| \leqslant \|\nabla\tau\|_\infty \cdot \|x - x'\|$$

$$|x - x' + t(\tau(x') - \tau(x))| \geqslant \frac{1}{2}\|x - x'\|,$$

where the last inequality follows from the reverse triangle inequality and the assumption $\|\nabla\tau\|_\infty \leqslant 1/2$. By Cauchy-Schwarz, and since $\|\nabla w(x)\|$ decreases with $\|x\|$, we have

$$\int |k(x, x')| dP(x') \leqslant \|\nabla\tau\|_\infty \int \|\nabla w((x - x')/2)\| \cdot \|x' - x\| \, dP(x')$$

$$\leqslant C_{\nabla w} \|\nabla\tau\|_\infty.$$

Similarly, we obtain $\int |k(x, x')| dP(x) \leqslant C_{\nabla w} \|\nabla\tau\|_\infty$. Then, Schur's test (Lemma 12) yields

$$\|A_\tau - A\| \leqslant C_{\nabla w} \|\nabla\tau\|_\infty. \tag{31}$$

**Bound on $\|D_{W_\tau}^{-1/2} - D^{-1/2}\|$.** Define $d = d_{W,P}$ and $d_\tau = d_{W_\tau, P}$. The operator $D_{W_\tau}^{-1/2} - D^{-1/2}$ is diagonal with elements $d_\tau(x)^{-1/2} - d(x)^{-1/2}$, such that $\|D_{W_\tau}^{-1/2} - D^{-1/2}\| \leqslant \left\| d_\tau^{-1/2} - d^{-1/2} \right\|_\infty$. Note that we have

$$|d_\tau(x) - d(x)| = \left| \int k(x, x') dP(x') \right| \leqslant C_{\nabla w} \|\nabla\tau\|_\infty,$$

Then as in the proof of Lemma 5 we have $\left\| d_\tau^{-1/2} - d^{-1/2} \right\|_\infty \leqslant c_{\min}^{-\frac{3}{2}} \|d_\tau - d\|_\infty \leqslant C_{\nabla w} c_{\min}^{-\frac{3}{2}} \|\nabla\tau\|_\infty$.

**Final bound.** Note that by Schur's test, we have $\|A\| \leqslant C_w$, $\|A_\tau\| \leqslant C_w + C_{\nabla w} \|\nabla\tau\|_\infty$. Further, we have $\|D^{-1/2}\|, \|D_{W_\tau}^{-1/2}\| \leqslant c_{\min}^{-1/2}$. Plugging back into (29), we have the desired bound.

### E.2 Proof of Theorem 4 and 5 (deformation change to $P$)

In this case, we have two random graph models with distributions $P$ and $P_\tau$, while $W$ remains fixed. Depending on the case, the input $f$ will change or not.

**Translation-invariant case.** We are going to prove Theorem 4 with $C$ defined as (26) and $C'$ as (28). Using (25) and Prop 2, we obtain that

$$\mathcal{W}_2\left(\Phi_{W,P_\tau}(f')_\sharp P_\tau, \Phi_{W,P}(f)_\sharp P\right) \leqslant \|T_\tau \Phi_{W,P_\tau}(f') - \Phi_{W,P}(f)\| = \|\Phi_{W_\tau,P}(f'_\tau) - \Phi_{W,P}(f)\|$$

In the invariant case, we have similarly that $\left|\bar\Phi_{W,P_\tau}(f') - \bar\Phi_{W,P}(f)\right| = \left|\bar\Phi_{W,P_\tau}(f') - \bar\Phi_{W,P}(f)\right| \leqslant \|\Phi_{W_\tau,P}(f'_\tau) - \Phi_{W,P}(f)\|$. Using Lemma 2 and the computation of the previous section, we obtain

$$\|\Phi_{W_\tau,P}(f'_\tau) - \Phi_{W,P}(f)\| \leqslant C(C_W + C_{\nabla w}) \|f\| \|\nabla\tau\|_\infty + C' \|f'_\tau - f\|,$$

where $C$ is defined by (26) and $C'$ is defined by (28).

*Proof of Prop. 3.* When using degree functions as input, using the proof of Prop 2 and the computations of the previous sections we have

$$\|T_\tau d_{W,P_\tau} - d_{W,P}\| = \|d_{W_\tau,P} - d_{W,P}\| \leqslant C_{\nabla w} \|\nabla\tau\|_\infty$$

$\square$

**Non Translation-invariant case.** We are going to prove Theorem 5 with constants $C$ defined as (26) and $C'$ defined as (28). By Assumption (A2) we easily have the following:

$$C_{P_\tau}^{-1/2} \|f\|_{L^2(P_\tau)} \leqslant \|f\| \leqslant C_{P_\tau}^{1/2} \|f\|_{L^2(P_\tau)} \tag{32}$$

such that for any $k$ we have $\left\|\mathcal{L}_{P_\tau}^k\right\| \leqslant C_{P_\tau}$ by observing that

$$\|\mathcal{L}_{P_\tau}^k f\| \leqslant C_{P_\tau}^{1/2} \|\mathcal{L}_{P_\tau}^k f\|_{L^2(P_\tau)} \leqslant C_{P_\tau}^{1/2} \|f\|_{L^2(P_\tau)} \leqslant C_{P_\tau} \|f\|$$

We can then prove the following Lemma.

**Lemma 3** (Stability in terms of measure change). *We have*
$$\left\|\bar{\Phi}_{W,P_\tau}(f) - \bar{\Phi}_{W,P}(f)\right\| \leqslant C_{P_\tau}^2 C_1 \left\|f\right\| \left\|\mathcal{L}_{P_\tau} - \mathcal{L}\right\| + C_2 \left\|f\right\| \left\|U_\tau - U\right\| \tag{33}$$
*with $C_1$ is the same as* (27) *and $C_2 = \|\theta\| \prod_{\ell=0}^{M-1} H_2^{(\ell)}$.*

*Proof.* From a simple triangle inequality and the estimate $\|U_\tau\| \leqslant C_{P_\tau}$ since $|U_\tau f| = \left|\int f dP_\tau\right| = \left|\int f q_\tau dP\right| \leqslant C_{P_\tau} \|f\|$, we have
$$\begin{aligned}
\left\|\bar{\Phi}_{W,P_\tau}(f) - \bar{\Phi}_{W,P}(f)\right\| &= \left\|U_\tau \Phi_{W,P_\tau}(f) - U\Phi_{W,P}(f)\right\| \\
&\leqslant C_{P_\tau} \left\|\Phi_{W,P_\tau}(f) - \Phi_{W,P}(f)\right\| + \left\|U_\tau - U\right\| \left\|\Phi_{W,P}(f)\right\|
\end{aligned}$$

We must now bound the first term. Denoting $f^{(\ell)}$ and $(f^{(\ell)})'$ the functions at each layer, we have using Lemma 6, (4) and the fact that $\left\|\mathcal{L}_{P_\tau}^k\right\| \leqslant C_{P_\tau}$:

$$\begin{aligned}
\left\|f^{(\ell)} - (f^{(\ell)})'\right\| &\leqslant \sqrt{\sum_j \left\|\sum_{i=1}^{d_{\ell-1}} h_{ij}^{(\ell-1)}(\mathcal{L}) f_i^{(\ell-1)} - h_{ij}^{(\ell-1)}(\mathcal{L}_{P_\tau})(f_i^{(\ell-1)})'\right\|^2} \\
&\leqslant \sqrt{\sum_j \left\|\sum_{i=1}^{d_{\ell-1}} (h_{ij}^{(\ell-1)}(\mathcal{L}) - h_{ij}^{(\ell-1)}(\mathcal{L}_{P_\tau}))(f_i^{(\ell-1)})'\right\|^2} \\
&\quad + \sqrt{\sum_j \left\|\sum_{i=1}^{d_{\ell-1}} h_{ij}^{(\ell-1)}(\mathcal{L})(f_i^{(\ell-1)} - (f_i^{(\ell-1)})')\right\|^2} \\
&\leqslant C_{P_\tau} H_{\partial,2}^{(\ell-1)} \left\|\mathcal{L} - \mathcal{L}_{P_\tau}\right\| \left\|(f^{(\ell-1)})'\right\| + H_2^{(\ell-1)} \left\|f^{(\ell-1)} - (f^{(\ell-1)})'\right\|
\end{aligned}$$

Then, we use Lemma 8 and the fact that there is no bias to obtain
$$\left\|(f^{(\ell-1)})'\right\| \leqslant C_{P_\tau}^{1/2} \left\|(f^{(\ell-1)})'\right\|_{L^2(P_\tau)} \leqslant C_{P_\tau}^{1/2} \|f\|_{L^2(P_\tau)} \prod_{s=0}^{\ell-1} H_2^{(s)} \leqslant C_{P_\tau} \|f\| \prod_{s=0}^{\ell-1} H_2^{(s)}$$

an easy recursion gives the result. $\qquad\qquad \square$

We first bound $\|U_\tau - U\|$ easily, by
$$|U_\tau f - Uf| = \left|\int f(x)dP_\tau(x) - \int f(x)dP(x)\right| = \left|\int f(x)(q_\tau(x) - 1)dP(x)\right| \leqslant N_P(\tau) \|f\|$$

which is also true for multivariate functions.

We now bound $\|\mathcal{L}_{P_\tau} - \mathcal{L}\|$. If we denote $J_\tau$ the diagonal change of variables operator with elements $q_\tau(x)$, we may write
$$\begin{aligned}
\mathcal{L}_{P_\tau} - \mathcal{L} &= D_{P_\tau}^{-1/2} A D_{P_\tau}^{-1/2} J_\tau - D^{-1/2} A D^{-1/2} \\
&= (D_{P_\tau}^{-1/2} - D^{-1/2}) A D_{P_\tau}^{-1/2} J_\tau + D^{-1/2} A (D_{P_\tau}^{-1/2} - D^{-1/2}) J_\tau + D^{-1/2} A D^{-1/2}(J_\tau - \mathrm{Id})
\end{aligned}$$

The following estimates are easily obtained using Schur's test or by a pointwise supremum: $\|A\| \leqslant C_w$, $\|J_\tau\| \leqslant C_{P_\tau}$, $\|J_\tau - \mathrm{Id}\| \leqslant N_P(\tau)$ and $\|D^{-1/2}\|, \|D_{P_\tau}^{-1/2}\| \leqslant c_{\min}^{-1/2}$. It is left to bound and $\|D_{P_\tau}^{-1/2} - D^{-1/2}\|$. As before,
$$\left\|D_{P_\tau}^{-1/2} - D^{-1/2}\right\| \leqslant c_{\min}^{-3/2} \|d_\tau - d\|_\infty \,.$$
and
$$\begin{aligned}
|d_\tau(x) - d(x)| &= \left|\int W(x,x')dP_\tau(x') - \int W(x,x')dP(x')\right| \\
&= \left|\int W(x,x')(q_\tau(x') - 1)dP(x')\right| \leqslant C_w N_P(\tau)
\end{aligned}$$

### E.3 Proof of Proposition 4 (signal deformations to $f$)

This is just a triangle inequality, combined with the previous theorems and Prop 2:

$$\left\|\bar{\Phi}_{W,P}(T_\tau f) - \bar{\Phi}_{W,P}(f)\right\| \leqslant \left\|\bar{\Phi}_{W,P}(T_\tau f) - \bar{\Phi}_{W_\tau,P}(T_\tau f)\right\| + \left\|\bar{\Phi}_{W_\tau,P}(T_\tau f) - \bar{\Phi}_{W,P}(f)\right\|$$

$$\leqslant C(C_W + C_{\nabla w})\|\nabla \tau\|_\infty \|T_\tau f\| + \left\|\bar{\Phi}_{W,P_\tau}(f) - \bar{\Phi}_{W,P}(f)\right\|$$

$$\leqslant C(C_W + C_{\nabla w})C_{P_\tau}^{1/2}\|f\|\|\nabla\tau\|_\infty + (CC_{P_\tau}^3 C_W + C')\|f\|N_P(\tau)$$

where $C$ is given by (26) and $C'$ is given by (28).

## F Technical Lemma

### F.1 Concentration inequalities

**Lemma 4** (Chaining on non-normalized kernels). *Consider a kernel $W$ and a probability distribution $P$ satisfying (6), any function $f \in \mathcal{B}(\mathcal{X})$, and $x_1, \ldots, x_n$ drawn iid from $P$. Then, with probability at least $1 - \rho$,*

$$\left\|\frac{1}{n}\sum_i W(\cdot, x_i)f(x_i) - \int W(\cdot, x)f(x)dP(x)\right\|_\infty \lesssim \frac{\|f\|_\infty \left(c_{\text{Lip.}}\sqrt{d_x} + (c_{\max} + c_{\text{Lip.}})\sqrt{\log \frac{n_\mathcal{X}}{\rho}}\right)}{\sqrt{n}}$$

*Proof.* Without lost of generality, we do the proof for $\|f\|_\infty \leqslant 1$. For any $x \in \mathcal{X}$, define

$$Y_x = \frac{1}{n}\sum_i W(x, x_i)f(x_i) - \int W(x, x')f(x')dP(x')$$

Since $\|W(\cdot, x)f\|_\infty \leqslant c_{\max}$, for any fixed $x_0 \in \mathcal{X}$, by Hoeffding's inequality we have: with probability at least $1 - \rho$,

$$|Y_{x_0}| \lesssim \frac{c_{\max}\sqrt{\log(1/\rho)}}{\sqrt{n}}$$

Consider any $j \leqslant n_\mathcal{X}$. For any $x_0 \in \mathcal{X}_j$, we have

$$\sup_{x \in \mathcal{X}_j}|Y_x| \leqslant \sup_{x,x' \in \mathcal{X}_j}|Y_x - Y_{x'}| + |Y_{x_0}|$$

The second term is bounded by the inequality above. For the first term, we are going to use Dudley's inequality "tail bound" version [44, Thm 8.1.6]. We first need to check the sub-gaussian increments of the process $Y_x$. For any $x, x' \in \mathcal{X}_j$, we have

$$\|Y_x - Y_{x'}\|_{\psi_2} \lesssim \frac{1}{n}\left(\sum_{i=1}^n \|(W(x, x_i) - W(x', x_i))f(x_i) - (T_{W,P}f(x) - T_{W,P}f(x'))\|_{\psi_2}^2\right)^{\frac{1}{2}}$$

$$\lesssim \frac{1}{n}\left(\sum_{i=1}^n \|(W(x, x_i) - W(x', x_i))f(x_i)\|_{\psi_2}^2\right)^{\frac{1}{2}}$$

$$\lesssim \frac{1}{n}\left(\sum_{i=1}^n \|(W(x, \cdot) - W(x', \cdot))f(\cdot)\|_\infty^2\right)^{\frac{1}{2}}$$

$$\leqslant \frac{c_{\text{Lip.}}}{\sqrt{n}}d(x, x')$$

where we have used, from [44], Prop. 2.6.1 for the first line, Lemma 2.6.8 for the second, Example 2.5.8 for the third, and the Lipschitz property of $W$ for the last.

Now, we apply Dudley's inequality [44, Thm 8.1.6] to obtain that with probability $1 - \rho$,

$$\sup_{x,x' \in \mathcal{X}_j}|Y_x - Y_{x'}| \lesssim \frac{c_{\text{Lip.}}}{\sqrt{n}}\left(\int_0^1 \sqrt{\log N(\mathcal{X}, d, \varepsilon)}d\varepsilon + \sqrt{\log(1/\rho)}\right)$$

$$\lesssim \frac{c_{\text{Lip.}}}{\sqrt{n}}\left(\sqrt{d_x} + \sqrt{\log(1/\rho)}\right)$$

Combining with the decomposition above and applying a union bound over the $\mathcal{X}_j$ yields the desired result. $\qquad\square$

**Lemma 5** (Chaining on normalized Laplacians). *Consider a kernel $W$ and a probability distribution $P$ satisfying (6), any function $f \in \mathcal{B}(\mathcal{X})$, and $x_1, \ldots, x_n$ drawn iid from $P$. Assume $n$ satisfies (17). Then with probability at least $1 - \rho$,*

$$\|(\mathcal{L}_X - \mathcal{L}_P)f\|_\infty \lesssim \frac{c_{\max}\|f\|_\infty D_{\mathcal{X}}(\rho)}{c_{\min}\sqrt{n}} \tag{34}$$

*where $D_{\mathcal{X}}(\rho) = \frac{1}{c_{\min}}\left(c_{\text{Lip.}}\sqrt{d_x} + (c_{\max} + c_{\text{Lip.}})\sqrt{\log \frac{n_{\mathcal{X}}}{\rho}}\right).$*

*Proof.* Again we assume $\|f\|_\infty \leqslant 1$ without lost of generality.

By Lemma 4 with $f = 1$ and (17), with probability $1 - \rho/2$ we have

$$\|d_X - d_P\|_\infty \lesssim \varepsilon_d \overset{\text{def.}}{=} \frac{c_{\text{Lip.}}\sqrt{d_x \log C_{\mathcal{X}}} + (c_{\max} + c_{\text{Lip.}})\sqrt{\log \frac{n_{\mathcal{X}}}{\rho}}}{\sqrt{n}} \leqslant \frac{c_{\min}}{2}$$

and in particular $d_X \geqslant c_{\min}/2$. In this case, for all $x$, we have

$$\left|\frac{1}{\sqrt{d_X(x)}} - \frac{1}{\sqrt{d_P(x)}}\right| \leqslant \frac{|d_P(x) - d_X(x)|}{\sqrt{d_X(x)d_P(x)}(\sqrt{d_X(x)} + \sqrt{d_P(x)})} \lesssim \frac{\varepsilon_d}{c_{\min}^{3/2}}$$

and for all $x, y$,

$$\left|\frac{1}{\sqrt{d_X(y)d_X(x)}} - \frac{1}{\sqrt{d_P(y)d_P(x)}}\right| \leqslant \frac{1}{\sqrt{d_X(y)}}\left|\frac{1}{\sqrt{d_X(x)}} - \frac{1}{\sqrt{d_P(x)}}\right|$$

$$+ \frac{1}{\sqrt{d_P(x)}}\left|\frac{1}{\sqrt{d_X(y)}} - \frac{1}{\sqrt{d_P(y)}}\right| \lesssim \frac{\varepsilon_d}{c_{\min}^2}$$

Now, define $\bar{W}(x,y) \overset{\text{def.}}{=} \frac{W(x,y)}{\sqrt{d_P(x)d_P(y)}}$. For all $j \leqslant n_{\mathcal{X}}$ and $x, x' \in \mathcal{X}_j$ and $y \in \mathcal{X}$, we have $\|\bar{W}(\cdot, y)\|_\infty \leqslant \frac{c_{\max}}{c_{\min}}$ and

$$\left|\bar{W}(x,y) - \bar{W}(x',y)\right| \leqslant \frac{|W(x,y)|}{\sqrt{d_P(y)}}\left|\frac{1}{\sqrt{d_P(x)}} - \frac{1}{\sqrt{d_P(x')}}\right| + \frac{1}{\sqrt{d_P(x')d_P(y)}}|W(x,y) - W(x',y)|$$

$$\lesssim \frac{c_{\text{Lip.}}c_{\max}}{c_{\min}^2}d(x,x')$$

Hence by applying Lemma 4 we obtain that with probability $1 - \rho/2$,

$$\left\|\frac{1}{n}\sum_i \bar{W}(\cdot, x_i)f(x_i) - \int \bar{W}(\cdot, x)f(x)dP(x)\right\|_\infty$$

$$\lesssim \varepsilon_W \overset{\text{def.}}{=} \frac{\frac{c_{\max}}{c_{\min}}\left(\frac{c_{\text{Lip.}}}{c_{\min}}\sqrt{d_x \log C_{\mathcal{X}}} + \left(1 + \frac{c_{\text{Lip.}}}{c_{\min}}\right)\sqrt{\log \frac{n_{\mathcal{X}}}{\rho}}\right)}{\sqrt{n}}$$

We can now conclude, observing that

$$\|(\mathcal{L}_X - \mathcal{L}_P)f\|_\infty = \left\|\frac{1}{n}\sum_i \frac{W(\cdot, x_i)}{\sqrt{d_X(\cdot)d_X(x_i)}}f(x_i) - \int \frac{W(\cdot, x)}{\sqrt{d_P(\cdot)d_P(x)}}f(x)dP(x)\right\|_\infty$$

$$\leqslant \sup_x \frac{1}{n}\sum_i |W(x, x_i)f(x_i)|\left|\frac{1}{\sqrt{d_X(x)d_X(x_i)}} - \frac{1}{\sqrt{d_P(x)d_P(x_i)}}\right|$$

$$+ \left\|\frac{1}{n}\sum_i \bar{W}(\cdot, x_i)f(x_i) - \int \bar{W}(\cdot, x)f(x)dP(x)\right\|_\infty$$

$$\lesssim \frac{c_{\max}}{c_{\min}^2}\varepsilon_d + \varepsilon_W \lesssim \frac{c_{\max}}{c_{\min}^2}\varepsilon_d$$

$\qquad\square$

### F.2   Misc. bounds

**Lemma 6** (Operator norms of filters). *Let $(E, \|\cdot\|_E)$ be a Banach space and $(\mathcal{H}, \|\cdot\|_{\mathcal{H}})$ be a separable Hilbert space. Let $L, L'$ be two bounded operators on $E$, and $S : E \to \mathcal{H}$ be a linear operator such that $\|S\|_{\mathcal{H} \to E} \leqslant 1$. For $1 \leqslant i \leqslant d$ and $1 \leqslant j \leqslant d'$, let $h_{ij} = \sum_k \beta_{ijk} \lambda^k$ be a collection of analytic filters, with $B_k = (\beta_{ijk})_{ji} \in \mathbb{R}^{d' \times d}$ the matrix of order-$k$ coefficients, with operator norm $\|B_k\|$. Let $x_1, \ldots, x_d \in E$ be a collection of points. Then:*

$$\sqrt{\sum_j \left\| S \sum_i h_{ij}(L) x_i \right\|_{\mathcal{H}}^2} \leqslant \left( \sum_k \|B_k\| \|L^k\| \right) \sqrt{\sum_i \|x_i\|_E^2}$$

*and*

$$\sqrt{\sum_j \left\| S \sum_i (h_{ij}(L) - h_{ij}(L')) x_i \right\|_{\mathcal{H}}^2} \leqslant \sum_k \|B_k\| \sqrt{\sum_i \left( \sum_{\ell=0}^{k-1} \|L^\ell\| \|(L - L')(L')^{k-1-\ell} x_i\|_E \right)^2}$$

*When $\mathcal{H}$ is only a Banach space, the same results hold with $B_{k,|\cdot|} = (|\beta_{ijk}|)_{ji}$ instead of $B_k$.*

*Proof.* Let $\{e_\ell\}_{l \geqslant 1}$ be an orthonormal basis for $\mathcal{H}$. For all $i, k$, decompose $SL^k x_i = \sum_\ell b_{ik\ell} e_\ell$. We have

$$\sqrt{\sum_j \left\| S \sum_i h_{ij}(L) x_i \right\|_{\mathcal{H}}^2} \leqslant \sqrt{\sum_j \left\| \sum_{ik} \beta_{ijk} SL^k x_i \right\|_{\mathcal{H}}^2} \leqslant \sum_k \sqrt{\sum_j \left\| \sum_{i\ell} \beta_{ijk} b_{ik\ell} e_\ell \right\|_{\mathcal{H}}^2}$$

$$\leqslant \sum_k \sqrt{\sum_\ell \sum_j \left( \sum_i \beta_{ijk} b_{ik\ell} \right)^2}$$

$$\leqslant \sum_k \sqrt{\|B_k\|^2 \sum_{i\ell} b_{ik\ell}^2} = \sum_k \|B_k\| \sqrt{\sum_i \|SL^k x_i\|_{\mathcal{H}}^2}$$

$$\leqslant \left( \sum_k \|B_k\| C^k \right) \sqrt{\sum_i \|x_i\|_E^2}$$

The proof of the second claim is obtained in the same way by decomposing $S(L^k - (L')^k) x_i$ in $\mathcal{H}$ and using at the last step:

$$\left\| (L^k - (L')^k) x \right\|_E = \left\| \sum_{\ell=0}^{k-1} L^\ell (L - L')(L')^{k-1-\ell} x \right\|_E \leqslant \sum_{\ell=0}^{k-1} C_\ell \left\| (L - L')(L')^{k-1-\ell} x \right\|_E$$

Finally, when $\mathcal{H}$ is not a Hilbert space, we directly use

$$\sqrt{\sum_j \left\| S \sum_i h_{ij}(L) x_i \right\|_{\mathcal{H}}^2} \leqslant \sqrt{\sum_j \left( \sum_{ik} |\beta_{ijk}| C^k \|x_i\|_E \right)^2} \leqslant \sum_k C^k \sqrt{\sum_j \left( \sum_i |\beta_{ijk}| \|x_i\|_E \right)^2}$$

$$\leqslant \left( \sum_k \|B_{k,|\cdot|}\| C^k \right) \sqrt{\sum_i \|x_i\|_E^2}$$

$\square$

**Lemma 7** (Lipschitz property of discrete GCNs). *Let $G_1 = (A, Z_1)$ and $G_2 = (A, Z_2)$ be two graphs with the same structure, and a GCN $\Phi$. Denote by $Z_r^{(M)}$ the signal at the last layer when applying $\Phi$ to $G_r$. We have*

$$\left\| Z_1^{(M)} - Z_2^{(M)} \right\|_F \leqslant \left( \prod_{\ell=0}^{M-1} H_2^{(\ell)} \right) \|Z_1 - Z_2\|_F$$

*Proof.* For $j \leqslant d_\ell$, using Lemma 6 and (4) we write

$$\left\|Z_1^{(M)} - Z_2^{(M)}\right\|_F \leqslant \sqrt{\sum_j \left\|\sum_{i=1}^{d_{M-1}} h_{ij}^{(M-1)}(L)(z_{1,i}^{(M-1)} - z_{2,i}^{(M-1)})\right\|^2}$$

$$\leqslant H_2^{(M-1)}\left\|Z_1^{(M-1)} - Z_2^{(M-1)}\right\|_F$$

An easy recursion gives the result. $\qquad\square$

**Lemma 8** (Bound on c-GCNs). *Apply a c-GCN to a random graph model $\Gamma = (P, W, f)$. Denote by $f^{(\ell)}$ the function at each layer. Then we have*

$$\left\|f^{(\ell)}\right\|_* \leqslant \|f\|_* \prod_{s=0}^{\ell-1} H_*^{(s)} + \sum_{s=0}^{\ell-1} \left\|b^{(s)}\right\| \prod_{p=s+1}^{\ell-1} H_*^{(p)} \tag{35}$$

*where $*$ indicates $L^2(P)$ or $\infty$.*

*Proof.* For $j \leqslant d_\ell$, using Lemma 6 and (4) we write

$$\left\|f^{(\ell)}\right\|_* \leqslant \sqrt{\sum_j \left\|\sum_{i=1}^{d_{\ell-1}} h_{ij}^{(\ell-1)}(\mathcal{L}_{W,P})f_i^{(\ell-1)} + b_j^{(\ell-1)}\right\|_*^2}$$

$$\leqslant \sqrt{\sum_j \left\|\sum_{i=1}^{d_{\ell-1}} h_{ij}^{(\ell-1)}(\mathcal{L}_{W,P})f_i^{(\ell-1)}\right\|_*^2} + \left\|b^{(\ell-1)}\right\|$$

$$\leqslant H_*^{(\ell-1)}\left\|f^{(\ell-1)}\right\|_* + \left\|b^{(\ell-1)}\right\|$$

An easy recursion gives the result. $\qquad\square$

**Lemma 9** (Piecewise Lipschitz property of c-GCNs). *Let $\Gamma$ be a random graph model. Assume that $f$ is piecewise $(c_f, n_f)$-Lipschitz. Then, $\Phi_{W,P}(f)$ is piecewise $(C, n_f n_\mathcal{X})$-Lipschitz with*

$$C = \|\theta\| \left(c_f \prod_{\ell=0}^{M-1} \left\|B_0^{(\ell)}\right\| + \frac{c_{\text{Lip}}.c_{\max}}{c_{\min}^2} \sum_{\ell=0}^{M-1} H_2^{(\ell)} \left\|f^{(\ell)}\right\|_{L^2(P)} \prod_{s=0}^{\ell-1} \left\|B_0^{(s)}\right\|\right) \tag{36}$$

*where $\left\|f^{(\ell)}\right\|_{L^2(P)}$ can be bounded by Lemma 8.*

*Proof.* Define the partition $\mathcal{X}_1', \ldots, \mathcal{X}_{n_f}'$ on which $f$ is Lipschitz, and take $x, x' \in \mathcal{X}_i \cap \mathcal{X}_j'$ for some $i, j$.

Using the same strategy as in the proof of Lemma 6, we have

$$\left\|f^{(M)}(x) - f^{(M)}(x')\right\| \leqslant \sqrt{\sum_j \left|\sum_i h_{ij}^{(M-1)}(\mathcal{L}_P)f_i^{(M-1)}(x) - h_{ij}^{(M-1)}(\mathcal{L}_P)f_i^{(M-1)}(x')\right|^2}$$

$$\leqslant \sum_k \left\|B_k^{(M-1)}\right\| \sqrt{\sum_i \left|\mathcal{L}_P^k f_i^{(M-1)}(x) - \mathcal{L}_P^k f_i^{(M-1)}(x')\right|^2}$$

Define $\bar{W}(x,y) = \frac{W(x,y)}{\sqrt{d_P(x)d_P(y)}}$. As we have seen in the proof of Lemma 5, $\bar{W}$ is piecewise $\frac{c_{\text{Lip}}.c_{\max}}{c_{\min}^2}$-Lipschitz on the $\mathcal{X}_i$. So, for $k \geqslant 1$, for $x, x' \in \mathcal{X}_i$ by Schwartz inequality we have

$$\left|\mathcal{L}_P^k f_i^{(M-1)}(x) - \mathcal{L}_P^k f_i^{(M-1)}(x')\right| \leqslant \left\|\mathcal{L}_P^{k-1} f_i^{(M-1)}\right\|_{L^2(P)} \frac{c_{\text{Lip}}.c_{\max}}{c_{\min}^2} d(x, x')$$

$$\leqslant \left\|f_i^{(M-1)}\right\|_{L^2(P)} \frac{c_{\text{Lip}}.c_{\max}}{c_{\min}^2} d(x, x')$$

And thus

$$\left\| f^{(M)}(x) - f^{(M)}(x') \right\| \leqslant \left\| B_0^{(M-1)} \right\| \left\| f^{(M)}(x) - f^{(M)}(x') \right\|$$
$$+ H_2^{(M-1)} \left\| f^{(M-1)} \right\|_{L^2(P)} \frac{c_{\text{Lip.}} c_{\max}}{c_{\min}^2} d(x, x')$$

A recursion gives the result, with Lemma 8. $\qquad\square$

## G   Third-party results

**Lemma 10** (Hoeffding's inequality)**.** *Let $X_1, \ldots, X_n \in \mathbb{R}$ be independent random variables such that $a \leqslant X_i \leqslant b$ almost surely. Then we have*

$$\mathbb{P}\left( \left| \frac{1}{n} \sum_i (X_i - \mathbb{E}X_i) \right| \geqslant \varepsilon \right) \leqslant 2 \exp\left( -\frac{2\varepsilon^2 n}{(b-a)^2} \right) \tag{37}$$

**Lemma 11** (Generalized Hoeffding's inequality [41])**.** *Let $\mathcal{H}$ be a separable Hilbert space and $\xi_1, \ldots, \xi_n \in \mathcal{H}$ be independent zero-mean random variables such that $\|\xi_i\| \leqslant C$ almost surely. Then with probability at least $1 - \rho$ we have*

$$\left\| \frac{1}{n} \sum_i \xi_i \right\| \leqslant \frac{C\sqrt{2\log(2/\rho)}}{\sqrt{n}} \tag{38}$$

**Theorem 6** (Spectral concentration of normalized Laplacian [25, Theorem 4])**.** *Let $A$ be an adjacency matrix of a graph drawn with independent edges $a_{ij} \sim \text{Ber}(\alpha_n p_{ij})$, where $p_{ij} \leqslant c_{\max}$ and for all $i$, $\frac{1}{n} \sum_j p_{ij} \geqslant c_{\min} > 0$. Denote by $P$ the $n \times n$ matrix containing the $p_{ij}$. There is a universal constant $C$ such that:*

$$\mathbb{P}\left( \|L(A) - L(P)\| \geqslant \frac{C(1+c)c_{\max}}{c_{\min}^2 \sqrt{n\alpha_n}} \right) \leqslant e^{-\left( \frac{3c^2}{12+4c} - \log(14) \right)n} + e^{-\frac{3c^2}{12+4c} n\alpha_n + \log(n)}$$
$$+ e^{-\frac{3c_{\min}^2 n\alpha_n}{25 c_{\max}} + \log(n)} + n^{-\frac{c}{4}+6} \tag{39}$$

**Theorem 7** (Wasserstein convergence [46])**.** *Let $(\mathcal{X}, d)$ be a compact metric space with $\text{diam}(\mathcal{X}) \leqslant B$ and $N_\varepsilon(\mathcal{X}) \leqslant (B/\varepsilon)^{d_x}$. Let $P$ be a probability distribution on $\mathcal{X}$, and $x_1, \ldots, x_n$ drawn iid from $P$. With probability $1 - \rho$,*

$$\mathcal{W}_2(\hat{P}, P) \lesssim B\left( n^{-\frac{1}{d_x}} + \left( 27^{\frac{d_x}{4}} + \log(1/\rho)^{\frac{1}{4}} \right) n^{-\frac{1}{4}} \right) \tag{40}$$

*where $\hat{P} = n^{-1} \sum_i \delta_{x_i}$.*

*Proof.* The result is obtained by combining Prop. 5 and Prop. 20 in [46] with $\varepsilon' = 1$, with the assumed simplified expression for the covering numbers of $\mathcal{X}$ and a rescaling of the metric such that $B$ disappears from the covering numbers expression. $\qquad\square$

**Lemma 12** (Schur's test)**.** *Let $T$ be the integral operator defined by*

$$Tf(x) = \int k(x, x')f(x')d\mu(x').$$

*If the kernel $k$ satisfies*

$$\sup_x \int |k(x, x')| d\mu(x') \leqslant C \quad \text{and} \quad \sup_{x'} \int |k(x, x')| d\mu(x) \leqslant C,$$

*then $T$ is bounded in $L^2(\mu)$, with $\|T\|_{L^2(\mu)} \leqslant C$.*