[Reviews · NeurIPS 2020]

Review 1

Summary and Contributions: This paper generalizes GCN on (discrete) graphs to a continuous version where the model operates on the continuous counterpart of graph signals. Authors show that under a certain class of random graph generative models, GCN converges to its continuous counterpart. Moreover, authors also provide stability results to show that both GCN and its continuous version are stable under the perturbation to the underlying random graph models. ---------- I read the response from the authors and other review comments. This paper is clearly a solid contribution to the community. I'd like to keep my original rating.

Strengths: 1, This paper has made quite a few important theoretical contributions. In particular, the continuous version of GCN is novel to me which may act as a bridge between manifold learning and graph learning. 2, The stability analysis may also have important implications to a range of applications like adversarial attacks and the generalization ability of GCN. I did not fully understand all the proof so that I can not fully tell its correctness. On the other hand, the proof techniques used by authors are quite involved and non-trivial. 3, The class of random graph generative models considered by authors is quite general which includes some famous models and potentially many real-world graphs as special cases.

Weaknesses: 1, It would be great to add more discussion on the implications of these results to practices. For example, it has been claimed that “the continuous GCN allows us to overcome the difficulties of dealing with discrete notions such as isomorphisms...”. However, after reading the results, I do not have a clear picture of how the continuous GCN is superior since you still need to deal with the continuous counterpart of permutation (e.g., translation or rotation of measures) as well. Similarly, for the stability analysis, I can foresee its applications to adversarial attack, generalization and so on. Further discussion along these lines would be very helpful to a larger group of audiences. 2, The overall paper is very dense as a conference paper. There are quite an amount of assumptions scattered around the paper. I would suggest authors summarize and discuss the most restricted ones so that people can easily understand the scope of the results. For example, like you did in the stability analysis, the notations of assumptions A1 and A2 help quite a bit. 3, The variable “d” is heavily overloaded in, e.g., the hidden dimension of GCN, distance, and constant. It would be great to improve the notation. 4, In line 134, “diam(X)” is used without introduction. In line 193, “If we where to use the latter...” is grammatically incorrect.

Correctness: I did not fully understand all the proof so that I can not fully tell its correctness. Overall, the results look quite reasonable to me.

Clarity: The most parts of the paper are well written except some notations could be improved.

Relation to Prior Work: The related work is adequately discussed.

Reproducibility: Yes

Additional Feedback: On the graph generative model side, one thing to mention is that the decoder used in graph auto-encoder [1] can be seen as one instance of the proposed class of random graph models. The edge probabilities are independent conditioned on the node latent variables. However, as shown in [2], compared to alternatives like auto-regressive models, it performs poorly in capturing real-world graphs like proteins. Of course, this does not necessarily mean that the expressiveness of the class of models is limited since the performance also depends on the learning process of such graph generative models. That being said, extending the analysis to other (more expressive) classes of generative models in the future would be very interesting. [1] Kipf, T.N. and Welling, M., 2016. Variational graph auto-encoders. arXiv preprint arXiv:1611.07308. [2] Liao, R., Li, Y., Song, Y., Wang, S., Hamilton, W., Duvenaud, D.K., Urtasun, R. and Zemel, R., 2019. Efficient graph generation with graph recurrent attention networks. In Advances in Neural Information Processing Systems (pp. 4255-4265).


Review 2

Summary and Contributions: This paper studies point-wise convergence and some kind of stability of Graph Convolutional Networks (without bias and defined by means of the normalized Laplacian) in some relatively sparse "graphon + noiseless signal" model, where relatively sparse means that the node degree is as low as logarithmic size of the graph and graphon + noiseless signal is defined by (5). Their generative model assume the existence of some latent space (supposed metric compact) as promoted by the theory of W-random graphs. This model encompasses most random graph models (SBM, epsilon-graphs, Geometric graphs,...). Their contribution is two-fold: first they derive convergence of GCN to its continuous counterpart referred to as "c-GCN" (Theorem 1); then in Section 4 they study some model of graph deformation (in the limit as the graphe size tends to infinity for which c-GCN is stable. A simple but not elementary contribution concerns invariance and equivariance of GCN and c-GCN (Propositions 1 and 2). The paper contains lots of nice remarks and discussions on the given results.

Strengths: This paper contains two important theoretical contributions: convergence of GCN and study of stability under graph deformation. These two kind of results have shown to be of the utmost importance in signal processing (eg, scattering transforms or certain CNN architectures), and I believe that this paper is an important and significant contribution to NeurIPS.

Weaknesses: The author(s) give a theoretical upper bound and heuristically argue that the rates (n\alpha_n)^{1/2} in Theorem 1 and n^{1/d_z} in Theorem 2 should be optimal. I agree with their argument while it is not a proof and rely on known open questions in the literature. It would have been nice to have a numerical experiment witnessing the rate (n\alpha_n)^{1/2} in Theorem 1. I know that computing the continuous GCN is difficult but one might look at simple cases where this computation should be tractable. [In a revised version the authors provide interesting numerical experiments on convergence]

Correctness: Claims, methods and discussion of the results are correct.

Clarity: I emphasize that it was a real pleasure to read this paper that presents in a synthetic and nice manner state-of-the-art results on this subject along with deep and interesting remarks on the results.

Relation to Prior Work: Short descriptions of the proofs are given and their positioning from previous contributions is clearly discussed.

Reproducibility: Yes

Additional Feedback: I think that this paper is influential and I recommend it for NeurIPS 2020. I have read the authors feedback, the other reviews and I keep my overall score to 9.


Review 3

Summary and Contributions: This paper aims to theoretically analyzes the convergence and stability of GCNs on large random graphs by defining intuitive notions of deformations and stability in the continuous space. To this end, the authors derive non-asymptotic convergence bounds on relatively sparse random graphs with non-smooth kernels, and introduce a Wasserstein-type stability bounds on equivariant c-GCNs.

Strengths: 1. This paper establishes connections between discrete and continuous GCNs, and further proves the convergence of discrete GCN to its counterpart continuous GCN under certain assumptions. 2. This paper gives promising stability properties and proofs of GCNs under some assumptions by relating the discrete and continuous GCNs. 3. This paper is theoretical and well written, and the corresponding proof seems to be correct.

Weaknesses: 1. The main concern is the lack of experimental results. This paper focuses on convergence and stability of large graphs, however, there is no simulation experiment to support the proposed theory, e.g. evaluating the convergence of GNN on real/simulation data. 2. The assumptions used in this paper are questionable, especially those in line 144 and 261. The authors must give clear illustrations to demonstrate that these assumptions are valid and not far from the practical case.

Correctness: Sounds correct.

Clarity: Yes

Relation to Prior Work: Yes

Reproducibility: No

Additional Feedback: Experimental results are suggested to be added to well support the proposed theory. ============= After seeing other reviews and author response, I give a score “accept”.


Review 4

Summary and Contributions: This paper presents theoretical analysis of convergence and stability properties of GCNs on large random graphs. It introduces continuous GCNs (c-GCN) that act on a bounded, piecewise-Lipschitz function of unobserved latent node variables which are linked through a similarity kernel. It has two main contributions. Firstly, it studies notions of invariance and equivariance to isomorphism of random graph models, and give convergence results of discrete GCNs to c-GCNs for large graphs. Specifically, for the invariant case the authors claim that the output of both networks lie in the same output space. For the equivariant case, to show convergence, they measure the MSE between node level output of either network. For the continuous case, this is the final layer output function measured at the sampled latent variable. Main convergence results are theorized with a bound on this MSE that stems from standard concentration inequalities. These results are non-asymptotic and do not assume smoothness of the similarity kernel. Further, authors consider the noisy observation scenario and show that given the Lipschitz property, deviation of GCN output from c-GCN due to noise converges to the standard deviation of the noise. They also provide a technique that uses the graph laplacian to filter out the noise. Secondly, it analyzes the stability of GCNs to small deformation of the underlying random graph model. Here, convergence results become useful as they facilitate treatment of GCNs in a continuous world, thus allowing the authors to define intuitive notions of model deformations and characterize their stability. Specifically, they theorize stability bounds on deformations of similarity kernels, node distributions and signals on the nodes. In a novel contribution, for GCNs equivariant to permutation, the authors relate existing discrete notions of distance between graph signals to a Wasserstein-type metric between the corresponding continuous representations.

Strengths: Soundness of claims: Claims are theoretically grounded and the corresponding technical assumptions are clearly stated. Significance of contributions: The significance of the convergence result lies in allowing a continuous domain treatment of GCNs. As the output of discrete GCNs has been shown to converge to that of c-GCN, tasks like classification and regression can be delegated to the more tractable c-GCNs. This could open the door to studying expressivity of GCNs and robustness to adversarial noise from a continuos setting. Significance could have been improved by providing at least preliminary results in these directons. Novelty: The authors do a fair job positioning their contributions within the existing body of work. Novelty is sufficient to merit publication. Relevance: With the increasing interest on GNNs, fundamental work that can help us understand their behavior is relevant to the ML community. The proposed work is nothing but a first step in this direction, but the idea of allowing us to operate on a continuous space seems to be a promising direction.

Weaknesses: The exposition of the paper gets too heavy in some portions. Enhaincing the intutitive explanations behind the main theorems would benefit the readers and streamline the message. Also, there is no experiment with synthetic data or otherwise (not even in the supplementary material) to empirically quantify the tightness or the functional forms of the bounds found. I understand that this is a theoretical paper, hence I would not punish it excesively for this, but it would have been preferable to have some demonstration in this direction.

Correctness: The results seem to be correct.

Clarity: The paper is quite well-written except for sporadic syntactical errors which can be neglected and the comment made above in the "Weaknesses" item. Clear objectives are presented at the beginning of each section along with assumptions that are relevant to that section or rest of the paper.

Relation to Prior Work: Authors provide clear distinctions of their work with prior arts that present theoretical analyses separately on convergence of graph operators on large graphs and stability of GCNs to multiple deformations and metrics. Specifically, authors claim superiority of their convergence analysis based on three aspects that include non-asymptotic results, applicability to large sparse graphs and preclusion of smoothness requirements of similarity kernels. For stability analysis, the main distinction lies in the use of continuous setup that allows the use of geometric perturbations based on deformations of random graph models and to obtain deformation stability bounds that are similar to those on Euclidean domains.

Reproducibility: Yes

Additional Feedback: The authors rightly mention that there are many papers "studying the convergence of graph-related objects on large random graphs", thus, it is unavoidable to miss some references in the related work such as the use of graphons for control design (see, e.g., Peter Caines' work) or in game theory (see, e.g., Asu Ozdaglar's work). However, the missing work that seems quite relevant and can motivate follow-up study is the idea of computing centrality values at the graphon level (see "Centrality Measures for Graphons: Accounting for Uncertainty in Networks" by Avella-Medina and others). In the current paper, the authors mention using the degree as an artificial feature in featureless settings, where the continuous counterpart (degree function) is easy to define (as also defined in Avella et al.). However, in Avella et al. other centrality measures are extended to the continuous graphon case, immediately allowing the input of a diversity of features to the c-GCN and, most probably, the extension of results like the ones presented here on Proposition 3 to a wider set of input features. I have read the authors' response and the idea of adding a simple experimental validation of their theoretical results is appealing.

[Author Response · NeurIPS 2020]

We thank the reviewers for their insightful feedback. We first address a general concern on numerical experiments,
before replying to reviewer-specific comments.

**Numerical experiments (R2, R3, R4).**  A shared comment by most referees is the absence of numerical experiments.
While the paper is theoretical in nature, and we initially chose to postpone in-depth numerical experiments for future
work for space reasons, we agree that simple experiments would be helpful to illustrate the theory. We will therefore
include simple synthetic experiments in the final version as an illustration of our theoretical findings. As a preliminary
study, Figure 1 illustrates (left) the convergence of a permutation-invariant GCN with random weights on simple
random graph models for different sparsity levels, and (center-right) its stability for a particular simple deformation of
three-dimensional latent variables. More refined experiments of this type, as well as the code, will be included in the
updated version of the paper.

Figure 1: (left) Convergence of permutation-invariant GCNs to its continuous value (approximated using a very large $n$ and $\alpha = 1$) on synthetic data. (middle-right) Illustration of stability to deformations. A Gaussian kernel is used to generate graphs from 3D latent positions on a surface. The deformation adds a ramp with height $\tau$. The middle figures illustrate the difference in output of an equivariant GCN between several draws of the random graph, with the same or deformed latent variables. With a translation-invariant kernel, "flat" parts on both side of the ramp yield the same output, despite being at different heights.

**R1.** *About "implications [...] to practices" and the "superiority" of continuous GCN*: In this paper, our goal is to
provide a novel theoretical analysis which may help explain the behavior of classical GCNs on typical large graphs,
rather than to propose a new model: c-GCNs are only theoretical objects that characterize the limit of GCNs. While we
indeed hope that our analysis will help derive guidelines for practices (for instance we mention helping to choose the
input signal in the absence of node features, but this can go beyond), this is a bit out-of-scope of the present paper and
left for future work. Such guidelines will likely come from in-depth analyses of each particular random graph model to
which our analysis apply: community graphs, geometric graphs, etc. We will update the discussion in the final version
of the paper. Finally, when we say "overcome the difficulties of dealing with [...] isomorphisms", we only meant in the
theoretical analysis. We will rephrase this to make this point clearer.
*"The overall paper is very dense [...]."*: The manuscript will be updated to use a unified notation for all assumptions.
*Use of the symbol d, $diam(X)$ and grammar comment*: We agree and will correct this in the update in order to improve
clarity. We will rephrase line 193.
*Additional references*: We thank the reviewer for pointing out these works. GVAE indeed exploits classical latent-space
random graph models (which date back all the way to the Erdös-Rényi model in the 50's) in the context of VAE. In
future work, additional properties of c-GCNs, e.g. approximation power, could probably be derived in this case. We
will include the suggested references.
**R2.** *Optimality of the bounds*: As pointed out by the referee, it is still unknown whether the bound in $(n\alpha_n)^{-1/2}$ is
optimal, as moreover for the case where $\alpha_n$ is proportional to $\log n/n$, the convergence is extremely slow and difficult
to observe in experiments. For the Wasserstein convergence in $n^{-1/d}$, lower bounds do exist however (see Weed and
Bach's paper), even if some questions are still open. We will update the discussion.
**R3.** *Assumptions*: Concerning the main assumption, the random graph modelling, it is indeed a central question to
assess how relevant they are to model real data. This however has a very long history in the literature, but we will add
some key references. The convergence assumption in line 144 allows us to analyze filters of infinite order, but it is
always satisfied for filters of finite order as commonly used in practice (e.g. in ChebNet). For the Euclidean assumption
line 261, we think that this is a simple assumption allowing us to discuss deformations through the formalism of
diffeomorphism, and it is valid for instance when considering graph models arising from 3D meshes. Both assumptions
could be extended in future work to handle different scenarios, and we will update the discussion accordingly.
**R4.** *About the exposition*: We will do our best to improve the discussions of our results in order to provide more
intuition, and hope the experiments presented above will help clarify our message on convergence and stability.
*Additional references*: Thank you for the additional references. Centrality measures are a key concept for graphons and
are definitely related to some considerations in our paper, and we will add the references in the updated paper.

[Meta-Review · NeurIPS 2020]

This paper considers a continuous version of graph convolutional neural network and analyze the usual discrete GCN as a discrete approximation of the continuous one. Under some random graph generative models, the convergence rate of the discrete one to the continuous one is derived. Moreover, some stability results are given to show that the induced GCN is stable against perturbation of the underlying generative model. The analysis is interesting and the expositions are well written. This kind of continuous-to-discrete type analysis would facilitate further theoretical analysis to understand GCN in general. Therefore, this paper is worth publication in NeurIPS. I encourage the authors to include the numerical experiments given in the feedback at least in the supplementary material.